# HABITAT 3.0: A CO-HABITAT FOR HUMANS, AVATARS AND ROBOTS

**Xavi Puig**[*], **Eric Undersander**[*], **Andrew Szot**[*], **Mikael Dallaire Cote**[*],
**Tsung-Yen Yang**[*], **Ruslan Partsey**[*], **Ruta Desai**[*], **Alexander William Clegg**[*],
**Michal Hlavac, So Yeon Min, Vladimír Vondruš, Theophile Gervet,**
**Vincent-Pierre Berges, John M. Turner, Oleksandr Maksymets, Zsolt Kira,**
**Mrinal Kalakrishnan, Jitendra Malik, Devendra Singh Chaplot, Unnat Jain,**
**Dhruv Batra, Akshara Rai**[†], **Roozbeh Mottaghi**[†]

http://aihabitat.org/habitat3

## ABSTRACT

We present Habitat 3.0: a simulation platform for studying collaborative human-robot tasks in home environments. Habitat 3.0 offers contributions across three dimensions: (1) **Accurate humanoid**[1] **simulation**: addressing challenges in modeling complex deformable bodies and diversity in appearance and motion, all while ensuring high simulation speed. (2) **Human-in-the-loop infrastructure**: enabling real human interaction with simulated robots via mouse/keyboard or a VR interface, facilitating evaluation of robot policies with human input. (3) **Collaborative tasks**: studying two collaborative tasks, Social Navigation and Social Rearrangement. Social Navigation investigates a robot's ability to locate and follow humanoid avatars in unseen environments, whereas Social Rearrangement addresses collaboration between a humanoid and robot while rearranging a scene. These contributions allow us to study end-to-end learned and heuristic baselines for human-robot collaboration in-depth, as well as evaluate them with humans in the loop. Our experiments demonstrate that learned robot policies lead to efficient task completion when collaborating with unseen humanoid agents and human partners that might exhibit behaviors that the robot has not seen before. Additionally, we observe emergent behaviors during collaborative task execution, such as the robot yielding space when obstructing a humanoid agent, thereby allowing the effective completion of the task by the humanoid agent. Furthermore, our experiments using the human-in-the-loop tool demonstrate that our automated evaluation with humanoids can provide an indication of the relative ordering of different policies when evaluated with real human collaborators. Habitat 3.0 unlocks interesting new features in simulators for Embodied AI, and we hope it paves the way for a new frontier of embodied human-AI interaction capabilities. The following video provides an overview of the framework: https://tinyurl.com/msywvsnz.

## 1 INTRODUCTION

Today's embodied AI agents are largely hermits – existing within and navigating through virtual worlds as solitary occupants (Batra et al., 2020; Anderson et al., 2017; Zhu et al., 2017; Chen et al., 2020; Krantz et al., 2020; Ku et al., 2020; Wani et al., 2020; Xia et al., 2020; Deitke et al., 2020; Gervet et al., 2023)[2]. Even in virtual worlds that feature interactivity (Deitke et al., 2022; Szot et al., 2021; Gan et al., 2021; Mu et al., 2021), where agents manipulate and move objects, the underlying assumption is that changes to the environment occur solely due to the actions of this one agent. These scenarios have formed a solid foundation for embodied AI and have led to the development of new architectures (Chaplot et al., 2020; Brohan et al., 2023; Shah et al., 2023; Kotar et al., 2023), algorithms (Wijmans et al., 2019; Wortsman et al., 2019; Li et al., 2019; Huang et al., 2022), and

---

[*] core team, † project leads. Work done at FAIR, Meta.

[1] Throughout this paper, we use *avatar* or *humanoid* to refer to virtual people in simulation and *human* to refer to real people in the world.

[2] With a few exceptions such as Li et al. (2021a).

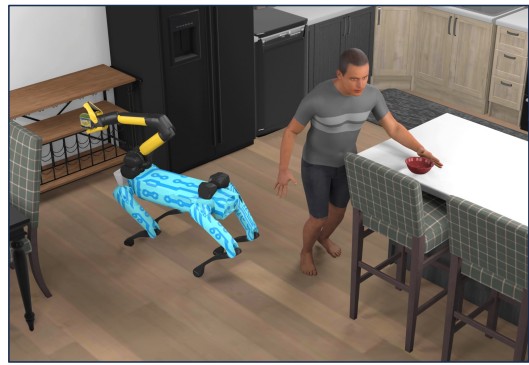 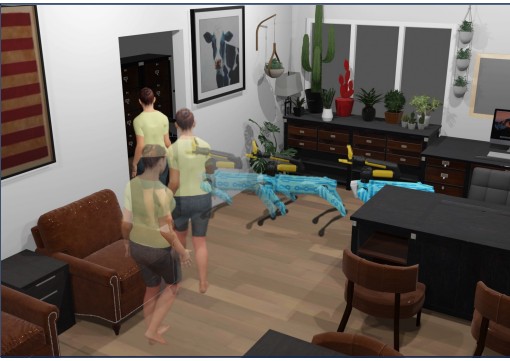

Figure 1: **Habitat 3.0:** An Embodied AI framework designed to facilitate simulation of human avatars and robotic agents within a wide array of indoor environments.

compelling simulation-to-reality results (Anderson et al., 2020; Kumar et al., 2021; Truong et al., 2022; Chebotar et al., 2018). However, they stand in stark contrast to the motivation behind the development of assistive robots, where the goal is to share the environment with human collaborators capable of modifying and dynamically altering it. We believe it is now time to more comprehensively study and develop *social embodied agents* that assist and cooperate with humans.

Consider the social robot shown in Fig. 1 – a Boston Dynamics Spot robot collaborating with a humanoid in a household setting, rearranging objects while cleaning up the house. To succeed in such tasks, the two agents need to efficiently divide the task based on their abilities and preferences while avoiding interference and respecting each other's space. For instance, the robot yields space to the humanoid by moving backward when passing through a door (Fig. 1, right).

Training and testing such social agents on hardware with real humans poses inherent challenges, including safety concerns, scalability limitations, substantial cost implications, and the complexity of establishing standardized benchmarking procedures. A simulation platform can overcome these challenges; however, the development of a collaborative human-robot simulation platform also comes with its own complexities. First, a major challenge is the modeling of deformable human bodies to synthesize realistic motion and appearance of the body parts. Second, to ensure the generalizability of human interaction models, it is crucial to incorporate diverse human behaviors and motions within the simulator. Finally, current state-of-the-art learning algorithms in Embodied AI often require a significant number of iterations to converge, leading to long training times. Hence, optimization techniques in human motion generation and rendering become crucial for achieving learning convergence within manageable wall-clock times.

In this paper, we introduce Habitat 3.0 — a simulator that supports both humanoid avatars and robots for the study of collaborative human-robot tasks in home-like environments.

**Human simulation –** Habitat 3.0 enables the following features for human simulation: (1) articulated skeletons with bones connected with rotational joints, facilitating fast collision testing, (2) a surface 'skin' mesh for high-fidelity rendering of the body, enhancing visual realism, (3) parameterized body models (by Pavlakos et al. (2019)) to generate realistic body shapes and poses, (4) a library of avatars made from 12 base models with multiple gender representations, body shapes, and appearances, (5) a motion and behavior generation policy, enabling the programmatic control of avatars for navigation, object interaction, and various other motions. Despite the emphasis on realism and diversity, the speed of our simulation remains comparable to scenarios involving non-humanoid agents (1190 FPS with a humanoid and a robot compared to 1345 FPS with two robots). This is achieved through optimization techniques that involve caching human motion data collected from diverse human shapes. This data is subsequently adapted to new environments using projection techniques, resulting in a simulation that is both fast and accurate, while respecting the geometry of the environment.

**Human-in-the-loop tool –** Beyond humanoid simulation, a pivotal feature of our system is a human-in-the-loop tool, an interactive interface, facilitating the evaluation of AI agents with real human collaborators. Through this tool, humans can collaborate with an autonomous robot using mouse and keyboard inputs or a virtual reality (VR) interface. This interactive setup enables us to evaluate the performance of learned AI agents within scenarios that closely resemble real-world interactions.

**Social tasks –** Aiming at reproducible and standardized benchmarking, we present two collaborative human-robot interaction tasks and a suite of baselines for each. We 'lift' the well-studied tasks of

*navigation* and *rearrangement* from the single-agent setting to a human-assistive setting. The first task, *Social Navigation*, involves the robot finding and following a humanoid while maintaining a safe distance. The second task, *Social Rearrangement*, involves a robot and a humanoid avatar working collaboratively to rearrange a set of objects from their initial locations to desired locations, using a series of pick-and-place actions (emulating the cleaning of a house). In social navigation, the robot's objective is independent of the humanoid's goal, while in social rearrangement the robot and the human must coordinate their actions to achieve a common goal as efficiently as possible. We evaluate these tasks within an automated setting, where both humanoid avatars and robots operate based on their learned/scripted policies. Furthermore, we evaluate social rearrangement in the human-in-the-loop framework using a subset of baselines, allowing real humans to control the humanoid avatar while the robot is controlled using a learned policy.

We conduct an in-depth study of learned and heuristic baselines on both tasks, with a focus on generalization to new scenes, layouts and collaboration partners. In social navigation, we find that end-to-end RL learns collaborative behaviors such as yielding space to the humanoid, ensuring their unobstructed movement. In social rearrangement, learned policies efficiently split the task between the robot and the humanoid, even for unseen collaborators, improving efficiency over the humanoid operating alone. These findings extend to our human-in-the-loop study, where social rearrangement baselines enhance humans' efficiency over performing the task alone.

In summary, we provide a platform for simulating humanoids and robots, thereby providing an environment for in-depth studies of social human-robot tasks. We hope that this platform will encourage greater engagement from the Embodied AI community in this important research direction. Our framework is open-sourced, for more details see Appendix A.

## 2   RELATED WORK

**Embodied AI Simulators.** Several simulators have been developed recently to support embodied tasks (Savva et al., 2019; Xia et al., 2018; Wani et al., 2020; Ramakrishnan et al., 2021; Deitke et al., 2020; Chen et al., 2020; James et al., 2020; Yu et al., 2019; Mu et al., 2021; Zhu et al., 2020; Kolve et al., 2017; Ehsani et al., 2021; Weihs et al., 2020; Szot et al., 2021; Shen et al., 2021; Gan et al., 2021; Xiang et al., 2020; Deitke et al., 2022). In contrast to these works, we simulate human movements and interactions in addition to robot simulation. iGibson 2.0 (Li et al., 2021b) simulates humanoids in the context of a social navigation task. However, humanoids are treated as rigid bodies that move on pre-defined tracks or rails. Building on iGibson 2.0, Srivastava et al. (2021) collect human demonstrations using a VR interface. VRKitchen (Gao et al., 2019) also provides such a VR interface to collect demonstrations and a benchmark to evaluate agents. These simulators focus on human teleoperation rather than humanoid simulation. Habitat 3.0 supports both humanoid simulation and human teleoperation. Overcooked-AI (Carroll et al., 2019) is a cooking simulator that is designed for coordination with humans, but their environments are simplified grid-like 2D environments. VirtualHome (Puig et al., 2018; 2021) is a simulator which supports humanoid avatars interacting in 3D environments, but does not support interactions with robots. In contrast, we simulate and model both robot and humanoid agents together and allow interactions between them. SEAN (Tsoi et al., 2022) supports both humanoid and robot simulation in 3D environments. However, it is limited to 3 scenes and primarily focused on social navigation. In contrast, we support object interaction in a wide range of indoor environments. Moreover, our simulator is an order of magnitude faster than existing humanoid simulators (detailed comparison in Table 4 in the Appendix).

**Human-in-the-Loop Training & Evaluation.** Human-in-the-loop (HITL) training (Fan et al., 2022; Zhang et al., 2021; MacGlashan et al., 2017) and evaluation (Hu et al., 2020; Puig et al., 2021; Carroll et al., 2019; Puig et al., 2023) have been studied in many different contexts such as NLP, computer vision, and robotics. Such studies in the Embodied AI domain are challenging, since they require an environment where both users and agents can interact. Prior works have focused on simplified grid or symbolic state environments (Carroll et al., 2019; Puig et al., 2021). A few works (Ramrakhya et al., 2022; Padmakumar et al., 2021; Das et al., 2018) collect user data in 3D environments at scale, but focus on data collection and offline evaluation rather than interactive evaluation. We present a HITL framework capable of interactive evaluation and data-collection.

**Embodied Multi-agent Environments & Tasks.** Most multi-agent reinforcement learning (MARL) works operate in a low-dimensional state space (e.g., grid-like mazes, tabular tasks, or Markov games) (Jaderberg et al., 2019; Samvelyan et al., 2019; oai; Resnick et al., 2018; Suarez et al., 2019;

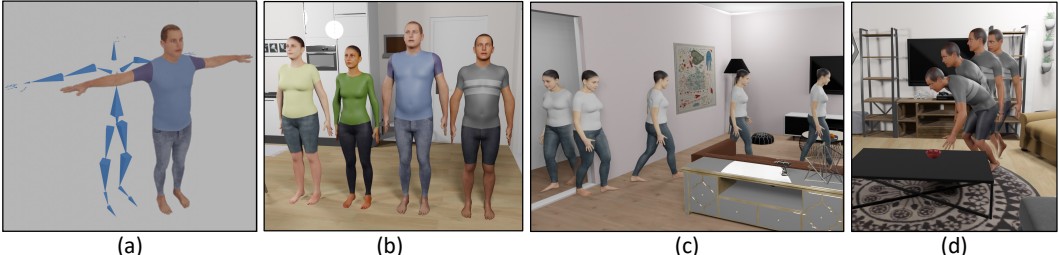

|      |      |      |      |
|------|------|------|------|
| (a)  | (b)  | (c)  | (d)  |

Figure 2: **Humanoid Avatars**. Visualization of the skeleton rig and skinned mesh (**a**). A subset of the sampled avatars featuring distinct genders, body shapes, and appearances (**b**). Simulation of realistic walking and reaching behaviors, which can adapt to different environment layouts (**c,d**).

Baker et al., 2020; Giles & Jim, 2002; Lazaridou et al., 2016; Foerster et al., 2016; Sukhbaatar et al., 2016; Mordatch & Abbeel, 2018). Different from these frameworks, Habitat 3.0 environments capture the realism necessary for household tasks. Closer to our work is prior work in visual MARL in 3D embodied settings: (Jain et al., 2019; 2020) introduce multiple agents in embodied simulation for moving furniture; Szot et al. (2023) study collaborative rearrangement in simulated home environments, where two robots perform tasks such as tidying up, and setting a dinner table. Thomason et al. (2020); Roman et al. (2020); Patel et al. (2021); Jain et al. (2021) study heterogeneous agents with different capabilities towards solving a shared task. In other visual MARL work such as (Chen et al., 2019; Juliani et al., 2018; Weihs et al., 2021b;a; Kurach et al., 2020) agents compete (e.g., hide-and-seek) in teams or individually. However, none of these prior works simulate humanoids. Wang et al. (2022) simulate humanoids and learn to generate human-robot collaborative behavior. In comparison, we scale each axis of simulation – scene and environment variations, and realism, fidelity and diversity of humanoid models.

## 3   HABITAT 3.0 FRAMEWORK

We first explain the intricacies of humanoid simulation and how we address key challenges such as efficiency, realism, and diversity. In particular, we focus on a caching strategy, the key element for achieving high speeds in appearance and motion generation. Then, we present our human-in-the-loop tool that enables evaluating robot policies with real humans in simulation.

### 3.1   HUMAN SIMULATION

To enable robots that learn to effectively interact with humans, we need to build fast, diverse, and realistic human models that can be deployed and used to train agents in simulation. We describe here how we design these humanoid agents, considering two components: their appearance and motion/behavior.

**Humanoid Appearance:** Following the traditional graphics approaches (Kavan & Žára, 2005; Kavan et al., 2007), we represent humanoids as an articulated skeleton, with bones connected via rotational joints (Fig. 2(a)). Additionally, a surface mesh is attached to the skeleton, and its vertices are updated based on the pose and shape of the skeleton using linear blend skinning (LBS) (Fig. 2(a)). The skeleton is used to represent poses and check for collisions with the environment, whereas the skinned mesh provides visual fidelity without affecting physics. The choice of an articulated skeleton for physics and a skinned mesh for appearance enhances the simulation speed without compromising fidelity. To generate realistic skeletons and meshes, we rely on SMPL-X (Pavlakos et al., 2019), a data-driven parametric human body model that provides a compact representation of 3D human shape and pose. In particular, SMPL-X represents a human's pose and shape into two sets of parameters $J, \beta$ for each gender. $J \in \mathbb{R}^{109}$ encodes a subject's pose via the hands, body and face joint rotations, and $\beta \in \mathbb{R}^{10}$ captures variations in body shape through the principle components of mesh vertex displacements. This allows us to create realistic body shapes by randomly sampling $\beta$. To encode body deformations in different poses, SMPL-X uses pose-dependent blend shapes, changing the relationship between the humanoid joints and skin mesh based on the pose. To enable fast humanoid simulation, we cache 4 male, 4 female and 4 neutral body shapes and store their corresponding skeleton and skinned meshes. This allows us to compute the model rig and blend shapes offline, only requiring LBS during simulation. Since the blend shapes are constant, we lose the ability to create pose-dependent skin deformations, resulting in a loss of some visual fidelity. However, this trade-off allows for the efficient loading and reposing of humanoid models. Fig. 2(b) and the supplemental video show some of the generated humanoids.

**Humanoid Motion and Behavior:** Our humanoid avatars are capable of realistic, long-range motion and behavior that can adapt to different tasks and environments. To achieve this, we design a hierarchical behavior model, in which a learned policy or planner executes a sequence of low-level skills to move the humanoid, generating long-range behaviors. Since our objective is to enable humanoid models to perform rearrangement and navigation tasks, we specifically consider two essential low-level skills: navigating around the environment and picking and placing objects[3].

For navigating, we first use a path planner to generate waypoints that reach the goal location without colliding with the environment. The humanoid then transitions between these waypoints by first rigidly rotating its base to face the next waypoint, and then moving forward. For animating the forward motion, we use a walking motion clip from the AMASS dataset (Mahmood et al., 2019), trim it to contain a single walking cycle (after both the left and right foot have moved) and play it cyclically until reaching the next waypoint (Fig. 2 (c)). For generating a picking motion, we use VPoser (Pavlakos et al., 2019) to pre-compute a set of humanoid poses for each body model, with one of the hands reaching a set of 3D positions around the human, while constraining the feet to be in a fixed location. We then store these poses offline, indexed by the 3D positions of the hand relative to the humanoid root. At evaluation time, we get humanoid poses for reaching arbitrary 3D points by interpolating over the closest stored poses, without requiring to query VPoser. This allows fast simulation of pick/place motions. A continuous picking and placing motion is obtained by drawing a 3D line from the current position of the hand to the target 3D position, and retrieving poses for intermediate points on the line. Once the hand reaches the object position, we kinematically attach or detach the object to or from the hand A resulting motion can be seen in Fig. 2(d). While this approach generates smooth motions for the range of positions computed on VPoser, it fails for positions out of this range, which happens when the hand is either too close or too far from the body.

**Benchmarking:** Our simulation design enables efficient humanoid simulation with minimal performance impact compared to simulating a robot. In our simulator, a robot operates at $245_{\pm 19}$ frames per second (FPS) in a single environment, while the humanoid achieves $188_{\pm 2}$ FPS. The difference in FPS is primarily due to the larger number of joints in the humanoid model, as compared to the robot. Removing skinning or motion capture data does not notably affect performance, implying that our simulation scales well to realistic visuals and animations. When using two agents, robot-robot achieves a frame rate of $150_{\pm 13}$, while robot-humanoid achieves $136_{\pm 8}$. Increasing the number of environments leads to substantial speed improvements, with robot-humanoid reaching $1191_{\pm 3}$ FPS across 16 environments on a single GPU. Further details are in Appendix F.1.

## 3.2 Human-in-the-Loop Infrastructure

We aim to study the generalization capabilities of our robot policies when interacting with collaborators that exhibit diverse behaviors, including real human collaborators. To facilitate human-robot interaction, we have developed a Human-in-the-Loop (HITL) *evaluation* platform. This tool allows human operators to control the humanoids within the simulated environment, using a mouse/keyboard or VR interface, enabling online human-robot interaction evaluations and data-collection. Below, we highlight some of key features:

**Ease-of-Development.** Our HITL tool leverages the infrastructure of AI Habitat simulator (Savva et al., 2019; Szot et al., 2021), enabling seamless integration and utilization of existing datasets and simulation capabilities. Further, we implement end-user logic in Python and provide convenient wrappers for the low-level simulation logic for ease of development.

**Portability to Other OS/Hardware Platforms.** We adopt a client-server architecture for our HITL tool that enables portability to other OS/hardware platforms. The server handles all logic (e.g., simulation, agent inference, and avatar keyboard controls), while the client handles only platform-specific rendering and input-device handling. This allows running the compute-heavy server component on a powerful machine while porting the client to other platforms, e.g., resource-constrained web browsers and VR devices. Hence, users can use this tool for a range of tasks: small-scale local evaluations using mouse and keyboard, large-scale data-collection on a browser, or high-fidelity human-robot interaction in VR.

**Replayability.** To support data collection and reproducibility, our platform provides functionalities for recording and replaying of HITL episodes. The playback functionality supports different levels of

---

[3]While our focus is on a restricted set of motions, the SMPL-X format allows us to add a wide range of motions, such as MoCap data, or the ones generated by motion generation models (e.g., Tevet et al. (2023)).

abstraction, ranging from high-level navigate/pick/place actions to precise trajectories of humanoid poses and rigid objects. Additionally, the platform provides the capability to re-render the entire episode from the viewpoints of different egocentric or exocentric cameras.

# 4 INTERACTIVE HUMAN-ROBOT TASKS

We study two tasks in Habitat 3.0: *social navigation* and *social rearrangement*. In social navigation, a robot must find and follow a humanoid, while maintaining a safe distance. In social rearrangement, a humanoid and a robot collaborate to move objects from their initial to target locations in a scene. Social navigation requires the robot to achieve its objective without interfering with the humanoid, while in social rearrangement, the two agents must collaborate to achieve a shared goal.

## 4.1 SOCIAL NAVIGATION: FIND & FOLLOW PEOPLE

**Task description.** An assistive robot should be able to execute commands such as "*Bring me my mug*", or "*Follow Alex and help him in collecting the dishes*", which require finding and following humans at a safe distance. With this in mind, we design a social navigation task (Francis et al., 2023), where a humanoid walks in a scene, and the robot must locate and follow the humanoid while maintaining a safety distance ranging from $1m$ to $2m$. Our task is different from prior work on social navigation such as Li et al. (2021a); Biswas et al. (2021) as they focus primarily on avoiding humans. It also differs from Luo et al. (2018), which focuses on following and tracking humans, but does not consider exploration to locate the humans, or constraints of the robot embodiment.

Fig. 3 (left) illustrates the social navigation task. The robot is placed in an unseen environment, and tasked with locating and following the humanoid, using a depth camera on its arm (arm depth), a binary humanoid detector that detects if the humanoid is in the frame (humanoid detector), and the humanoid's relative distance and heading to the robot (humanoid GPS), available to the robot at all times. This task is different from other navigation tasks such as image-goal (Zhu et al., 2017), point-goal (Anderson et al., 2018), and object-goal (Batra et al., 2020) navigation since it requires reasoning about a dynamic goal and dynamic obstacles. The robot must adapt its path based on the position of the humanoid and anticipate the humanoid's future motion to avoid colliding with it. The episode terminates if the robot collides with the humanoid or reaches maximum episode length. The humanoid follows the shortest path to a sequence of randomly sampled points in the environment.

**Metrics.** We define 3 metrics for this task: (1) *Finding Success* (S): Did the robot find the humanoid within the maximum episode steps (reach within 1-$2m$ while facing the humanoid)? (2) *Finding Success Weighted by Path Steps* (SPS): How efficient was the robot at finding the humanoid, relative to an oracle with ground-truth knowledge of the full humanoid trajectory and a map of the environment? Let $l$ be the minimum steps that such an oracle would take to find the humanoid, and $p$ be the agent's path steps, SPS $= S \cdot \frac{l}{\max(l,p)}$. (3) *Following Rate* (F): Ratio of steps that the robot maintains a distance of 1-$2m$ from the humanoid while facing towards it, relative to the maximum possible following steps. For an episode of maximum duration $E$, we assume that an oracle with ground-truth knowledge of humanoid path and the environment map can always follow the humanoid, once found. Thus, the following steps for such an oracle are $E - l$, where $l$ is the minimum number of steps needed to find the humanoid. Let $w$ denote the number of steps that the agent follows the humanoid, then following rate F $= \frac{w}{\max(E-l,w)}$. (4) *Collision Rate* (CR): Finally, we report the ratio of episodes that end in the agent colliding with the humanoid. More details about the metrics are in Appendix A.

**Baselines.** We compare two approaches in the social navigation task:

- **Heuristic Expert**: We create a heuristic baseline with access to the environment map that uses a shortest path planner to generate a path to the humanoid's current location. The heuristic expert follows the following logic: When the agent is farther than $1.5m$ from the humanoid, it uses the 'find' behavior, i.e., uses a path planner to approach the humanoid. If the humanoid is within $1.5m$, it uses a backup motion to avoid colliding with the humanoid.

- **End-to-end RL**: A 'sensors-to-action' recurrent neural network policy, trained using DDPPO (Wijmans et al., 2019). Inputs to the policy are egocentric arm depth, a humanoid detector, and a humanoid GPS, and outputs are velocity commands in the robot's local frame. This policy does not have access to a map of the environment. Architecture and training details are in Appendix A. We also provide ablations where we study the impact of the different sensors input to the policy.

| | S↑ | SPS↑ | F↑ | CR↓ |
|---|---|---|---|---|
| Heuristic Expert | 1.00 | 0.97 | 0.51 | 0.52 |
| End-to-end RL | $0.97_{\pm 0.00}$ | $0.65_{\pm 0.00}$ | $0.44_{\pm 0.01}$ | $0.51_{\pm 0.03}$ |
| - humanoid GPS | $0.76_{\pm 0.02}$ | $0.34_{\pm 0.01}$ | $0.29_{\pm 0.01}$ | $0.48_{\pm 0.03}$ |
| - humanoid detector | $0.98_{\pm 0.00}$ | $0.68_{\pm 0.00}$ | $0.37_{\pm 0.01}$ | $0.64_{\pm 0.05}$ |
| - arm depth | $0.94_{\pm 0.01}$ | $0.54_{\pm 0.01}$ | $0.19_{\pm 0.01}$ | $0.71_{\pm 0.08}$ |
| - arm depth + arm RGB | $0.96_{\pm 0.00}$ | $0.61_{\pm 0.01}$ | $0.38_{\pm 0.02}$ | $0.55_{\pm 0.04}$ |

Figure 3: **Social Navigation.** Overview of the Social Navigation task and sensors used (**left**). Baseline Results (**right**). The bottom rows show different variations of removing sensors.

**Scenes and Robot.** We incorporate the Habitat Synthetic Scenes Dataset (HSSD) (Khanna et al., 2023) in Habitat 3.0, and use the Boston Dynamics (BD) Spot robot. We use 37 train, 12 validation and 10 test scenes. For details on the robot and scenes, refer to Appendix D and E.

**Results.** Fig. 3 shows the performance of different baselines. The heuristic expert achieves 100% finding success (S) vs 97% for the RL policy. This is expected as the heuristic has privileged access to a map of the environment, and can compute the shortest path to the humanoid while the RL policy has to explore the environment. As a result, the heuristic expert finding SPS is also higher than the RL policy. However, even without access to a map, the RL policy learns to follow the humanoid by anticipating their motion, backing up to avoid collisions, and making way in narrow spaces, resulting in a similar success (S) and collision rate (CR) as the heuristic expert ($0.52$ for heuristic vs $0.51_{\pm 0.03}$ for RL policy), and a competitive SPS and following rate F ($0.51$ for heuristic expert vs $0.44_{\pm 0.01}$ for the RL policy). For qualitative results, refer to the supplementary video and Appendix C.1.

**Ablations.** We analyze the impact of individual sensors on the end-to-end RL policy's performance (bottom rows in Fig. 3). In general, the results can be grouped into two cases: *before* and *after* finding the humanoid. Before finding the humanoid, the humanoid GPS is the most important sensor since it enables agents to locate the humanoid. As a result, the baseline without the humanoid GPS has the lowest finding success and SPS. However, after finding the humanoid, arm perception (either RGB or depth) becomes important since the agent uses this information to avoid collision while following the humanoid. As a result, the baseline without arm cameras tends to have higher collision rate and lower following rate than the baselines with either a depth or RGB sensor.

## 4.2 SOCIAL REARRANGEMENT

**Task Description.** In this task (Fig. 4, left), the robot's goal is to efficiently assist a humanoid collaborator in rearranging two objects from known initial positions to designated goals. Object positions and goals are specified as 3D coordinates in the robot's start coordinate frame. We modify Social Rearrangement from Szot et al. (2023), generalizing it to heterogeneous agents (a humanoid and a robot) and dealing with more diverse, unseen environments and humanoid collaborators. The objects are spawned on open receptacles throughout an unseen house, with multiple rooms, and assigned a goal on another open receptacle within the house. The robot has access to its own egocentric depth cameras, proprioceptive state information (arm joint angles and base egomotion), and the humanoid's relative distance and heading. However, it does not have access to the humanoid's actions, intents, or complete states. Episodes end when all objects are placed in their desired locations or the maximum allowed steps are reached. During evaluation, the trained robot collaborates with new human or humanoid partners in diverse, unseen homes, as described later in this section.

**Metrics.** We evaluate all approaches on two metrics: (1) *Success Rate* (SR) at completing the task in an unseen home, with an unseen configuration of objects. An episode is considered successful (SR = 1) if both objects are placed at their target locations, within the maximum number of allowed episode steps. (2) *Relative Efficiency* (RE) at task completion, relative to the humanoid doing the task alone. For an episode with maximum allowed steps $E$, if the humanoid alone takes $L^{human}$ steps, and the humanoid-robot team takes $L^{joint}$ steps to finish the task, RE $= \frac{L^{human}}{\max(L^{joint}, E)}$. If the humanoid-robot team finishes the task in half the steps as humanoid alone, RE will be 200%.

**Policy architecture.** We adopt a two-layer policy architecture for all baselines, where a learned high-level policy selects a low-level skill from a pre-defined library of skills to execute based on observations. We consider 2 variants of low-level skills for the robot – *oracle* and *learned* low-level skills. Oracle skills use privileged information, i.e., a map of the environment for navigation and perfect, 'instantaneous' pick/place skills. Learned low-level skills are pre-trained and frozen, but are realistic and do not assume privileged information, resulting in more low-level failures. Refer to Appendix A for more details. All humanoid low-level skills are as described in Section 3.1.

**Baselines.** We study 3 different population-based approaches (Jaderberg et al., 2019) at the social rearrangement task, where a single high-level robot policy is trained to coordinate with collaborators within a 'population'. During training, for each episode, one collaborator is randomly selected from the population, and the robot is trained to collaborate with it. The robot policy is trained using end-to-end RL, using the same policy architecture, training parameters and scenes across all approaches. The approaches differ only in the training population.

- **Learn-Single**: In this baseline, we jointly learn a humanoid and a robot policy; the population consists of a single humanoid policy, resulting in low training collaborator diversity.
- **Plan-Pop$_p$**: We consider four different population sizes, with collaborators driven by privileged task planners. This results in four baselines (indexed by $p$), consisting of population sizes of 1 to 4. (1) $p = 1$ is a population of size 1, with a humanoid collaborator consistently rearranging the same object across all episodes. (2) $p = 2$ is a population of size 2, where each collaborator is responsible for one of the two objects. (3) $p = 3$ is a population of size 3, where collaborators rearrange either one or both objects. (4) $p = 4$ is a size 4 population where collaborators rearrange either one, both or none of the objects. Note that in all baselines, there is only one humanoid in the scene with the robot at a time, randomly sampled from the population. We train a different robot policy per population, and present results on all four baselines.
- **Learn-Pop**: In this baseline, instead of relying on a privileged planner to form the population, we learn a population of collaborators, following the approach of population-play (Jaderberg et al., 2019). We randomly initialize 8 humanoid policies, considering that different initializations may lead to diverse final behaviors. During training, each episode pairs a randomly chosen humanoid policy from this population with the robot, and both are jointly trained at the collaborative task. This baseline examines the efficacy of random initialization at learning populations.

**Evaluation population.** We use the same agent embodiment and train/evaluation dataset as in Social Navigation. However, social rearrangement evaluation also considers different collaborators, with different high-level behaviors. We create a population of 10 collaborators for zero-shot coordination (ZSC-pop-eval) by combining a subset of training collaborators from all baselines. Specifically, we collect 3 learned humanoid policies from the Learn-Single and Learn-Pop baselines, and 4 planner-based collaborators (one from each population $p$ in Plan-Pop$_p$). As a result, each baseline sees about 1/3 of the ZSC-eval collaborators during learning and needs to generalize to the remaining 2/3 of the population. We also evaluate each approach against its training population (train-pop-eval) but in unseen scenes and environment configurations. This lets us study the difference between collaborating with known collaborators (train-pop-eval) and unseen collaborators (ZSC-pop-eval).

**Results.** Fig. 4 shows the results of train-pop and ZSC-pop evaluation of the different baselines. Learn-Single and Plan-Pop$_1$ result in the highest success rate when evaluated against their training population. This is expected since both are trained with only one partner, and hence have the simplest training setting. However, their SR significantly drops during ZSC-pop-eval (98.50% → 50.9% and 91.2% → 50.4%). This is because these baselines are trained to coordinate with just a single partner, and hence have poor adaptability to partners that are different from their training partner. Amongst approaches trained with a population > 1, Learn-pop has the highest train-pop SR but poor ZSC-eval performance (92.2% → 48.5%), because of low emergent diversity in the learned population. In contrast, Plan-pop$_{2,3,4}$ have a smaller drop in performance between train-pop and ZSC-pop, signifying their ability to adapt to unseen partners. In particular, Plan-pop$_{3,4}$ perform similarly, with the highest ZSC-pop-eval SR of 71.7%. We observe similar trends in RE, where Learn-Single has the highest RE with train-pop, and a drop with ZSC-pop (159.2 → 106.0). Plan-pop$_{3,4}$ have a lower drop (105.49 → 101.99 for Plan-pop$_4$) but on average result in similar RE as Learn-Single and Plan-Pop$_1$ in ZSC-pop-eval. This is because on average over the 10 ZSC partners, Learn-Single significantly improves efficiency for some, while making others inefficient, as seen by its large variance. On the other hand, Plan-pop$_{3,4}$ are able to adapt and make most partners slightly more efficient, as shown by their lower variance, but on average perform similar to Learn-Single. For more details and additional analysis, refer to Appendix C.

**Ablations.** We conduct ablation experiments over inputs to the robot policy, trained with a Plan-Pop$_3$ population (Fig. 4 (right), bottom rows). We replace 'oracle' skills with non-privileged learned low-level skills without re-training the high-level policy. We find a notable drop in performance, stemming from low-level execution failures (77.79% → 41.96% in train-pop-eval SR), indicating that the performance can be improved by training the high-level policy with learned skills to better adapt to such failures. We also retrain our policies with RGB instead of depth, and do not observe a

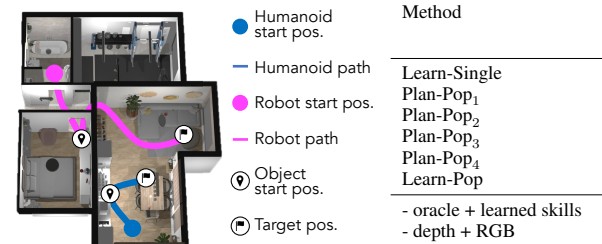

| Method | Train-pop-eval | | ZSC-pop-eval | |
|---|---|---|---|---|
| | SR↑ | RE↑ | SR↑ | RE↑ |
| Learn-Single | $98.50_{\pm0.48}$ | $159.2_{\pm1.0}$ | $50.94_{\pm39.55}$ | $106.02_{\pm34.32}$ |
| Plan-Pop$_1$ | $91.2_{\pm2.63}$ | $152.4_{\pm5.4}$ | $50.44_{\pm39.02}$ | $109.75_{\pm34.63}$ |
| Plan-Pop$_2$ | $66.89_{\pm1.47}$ | $110.06_{\pm6.83}$ | $70.23_{\pm7.02}$ | $102.13_{\pm11.10}$ |
| Plan-Pop$_3$ | $77.79_{\pm2.86}$ | $118.95_{\pm6.04}$ | $71.79_{\pm7.38}$ | $101.99_{\pm15.18}$ |
| Plan-Pop$_4$ | $72.42_{\pm1.32}$ | $105.49_{\pm1.7}$ | $71.32_{\pm6.47}$ | $103.53_{\pm9.8}$ |
| Learn-Pop | $92.20_{\pm2.21}$ | $135.32_{\pm3.43}$ | $48.52_{\pm35.51}$ | $99.80_{\pm31.02}$ |
| - oracle + learned skills | $41.09_{\pm21.5}$ | $79.62_{\pm1.76}$ | $21.44_{\pm18.16}$ | $76.45_{\pm9.23}$ |
| - depth + RGB | $76.70_{\pm3.15}$ | $110.04_{\pm3.05}$ | $70.89_{\pm8.18}$ | $100.16_{\pm14.79}$ |
| - Humanoid-GPS | $76.45_{\pm1.85}$ | $108.96_{\pm2.66}$ | $68.70_{\pm6.75}$ | $98.58_{\pm10.32}$ |

Figure 4: Overview of social rearrangement (**left**). Baseline results, averaged over 3 seeds (**right**).

| Method | CR ↓ | | TS ↓ | | RC ↑ | | RE ↑ |
|---|---|---|---|---|---|---|---|
| | Mean | 95% CI | Mean | 95% CI | Mean | 95% CI | |
| Solo | **0.0** | - | 1253.17 | [1020.90 1533.26] | - | - | 100.0 |
| Learn-Single | 0.12 | [0.19 0.08] | **936.60** | [762.50 1146.62] | 0.36 | [0.33 0.39] | **133.80** |
| Plan-Pop$_3$ | 0.13 | [0.20 0.08] | 1015.05 | [826.42 1242.60] | **0.44** | [0.41 0.46] | 123.46 |

Table 1: **Human-in-the-Loop Coordination Results.** We report estimated mean and 95% confidence intervals (CI) across 30 participants.

drop in performance. Removing the humanoid-GPS results in a slight drop in SR and RE, indicating that this sensor is useful for collaboration, especially in the ZSC setting, though not essential.

### 4.3 HUMAN-IN-THE-LOOP EVALUATION

We test trained robotic agents' ability to coordinate with real humans via our human-in-the-loop (HITL) tool across 30 participants. After a brief keyboard/mouse control training, we ask the participants to perform the social rearrangement task in the test scenes from our dataset. Particularly, the study operates under 3 conditions: performing the task alone (*solo*), paired with a robot operating with *Learn-Single*, or paired with a *Plan-Pop$_3$* agent. Each participant performs the task for 10 episodes per condition in one of the test scenes. We measure the collision rate (CR), task completion steps (TS), ratio of task completed by the robot (RC), and Relative Efficiency (RE) across all episodes. RE is the same as in Sec 4.2. CR is the ratio of episodes where the robot collides with the humanoid. RC is the proportion of objects that were rearranged by the robot per episode.

Both Plan-Pop$_3$ and Learn-Single improve RE to $123\%$ and $134\%$ respectively (Tab. 1). This shows that the robot makes the human more efficient than the human operating alone, even for completely unseen real human partners. Our analysis shows that pairwise difference of the estimated mean for TS between solo and Learn-Single, and solo and Plan-Pop$_3$ was significant, but the difference between estimated mean TS of Learn-Single and Plan-Pop$_3$ was not (details in Appendix G). Plan-Pop$_3$ leads to higher task offloading as measured by RC despite having lower RE than Learn-single.

In general, we observe that humans are more reactive to robot behavior than ZSC agents, which leads to a success rate of 1 across all episodes and high RE. For example, humans quickly adapt their plan based on their inferred goal of the robot, or move out of the way to let the robot pass. However, the relative order of average RE and RC between our automated and HITL evaluations holds, wherein Learn-Single makes the partner more efficient than Plan-Pop$_3$, but Plan-Pop$_3$ has higher RC than Learn-Single. This reveals a few interesting insights: (1) The automated evaluation pipeline can give an indication of the relative ordering of different approaches when evaluated with real human partners. (2) Our ZSC agents do not accurately capture the dynamics of human-robot interaction, and there is room for improvement. (3) Even approaches such as Learn-Single, which do not use a diverse training population, can enhance human efficiency compared to performing a task alone.

## 5 CONCLUSION

We introduce Habitat 3.0, an Embodied AI simulator designed to efficiently simulate humanoids and robots within rich and diverse indoor scenes. Habitat 3.0 supports a diverse range of appearances and motions for humanoid avatars, while ensuring realism and fast simulation speeds. In addition to humanoid simulation, we provide an infrastructure for human-in-the-loop (HITL) control of humanoid avatars, via a mouse/keyboard or VR interface. This interface allows us to collect real human-robot interaction data in simulation and evaluate robot policies with real humans. These capabilities allow us to study two collaborative tasks - social navigation and social rearrangement in both automated and human-in-the-loop evaluation settings. We observe emergent collaborative behaviors in our learned policies, such as safely following humanoids, or making them more efficient by splitting tasks. Our HITL analysis reveals avenues for enhancing social embodied agents, and we hope that Habitat 3.0 will accelerate future research in this domain.

## 6 ACKNOWLEDGEMENTS

We thank the reviewers for their helpful suggestions. We would also like to thank Manolis Savva and Angel Chang for their support in developing and curating the HSSD scenes used in this work. We are also grateful to Meshcapade GmbH for their support in incorporating and releasing the SMPL-X body models and sample motions to represent humanoids in Habitat.

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

APPENDIX

## A  IMPLEMENTATION DETAILS

We include the implementation details necessary for reproducing results for the tasks described in Section 4.   Code for reproducing experimental results can be found here: https://github.com/facebookresearch/habitat-lab/tree/v0.3.0/habitat-baselines#habitat-30-multi-agent-training.

### A.1  SOCIAL NAVIGATION

The robot uses neural network policies to find the humanoid, and the humanoid is scripted to navigate to random waypoints using a shortest path planner. We train all the end-to-end RL social navigation baselines using DD-PPO (Wijmans et al., 2019), distributing training across 4 NVIDIA A100 GPUs. Each GPU runs 24 parallel environments, and collects 128 steps for each update. We use a long short-term memory networks (LSTM) (Hochreiter & Schmidhuber, 1997) policy with ResNet18 as the visual backbone and two recurrent layers, resulting nearly 8517k parameters. We use a learning rate of $1 \times 10^{-4}$ and the maximum gradient norm of $0.2$. It takes about 200 million environment steps (roughly 4 days of training) to saturate. All baselines are trained with 3 different random seeds, and results are reported averaged across those seeds. Inspired by the reward design in training point-goal (Anderson et al., 2018), and object-goal policies (Batra et al., 2020), the social navigation reward is based on the distance to the humanoid at time $t$, and defined as follows:

$$r_t^{\text{distance}} = \begin{cases} \Delta(b_t, h_t) - \Delta(b_{t-1}, h_{t-1}), & \text{if } \Delta(b_t, h_t) \leq 1 \\ 2, & \text{if } 1 < \Delta(b_t, h_t) \leq 2 \\ \Delta(b_{t-1}, h_{t-1}) - \Delta(b_t, h_t), & \text{otherwise} \end{cases}$$

where $\Delta(b_t, h_t)$ is the geodesic distance between the robot location $b_t$ and the humanoid location $h_t$ at time $t$. The first condition encourages the robot to move away from the humanoid when it is closer than $1m$, ensuring that the agent maintains a safe distance from the humanoid. The second condition gives a constant reward for reaching within 1-2m, and the last condition rewards the robot to get closer to the humanoid.

To make sure the robot can face toward the humanoid, we further add an orientation reward when the robot is approaching the humanoid:

$$r_t^{\text{orientation}} = \begin{cases} (h_t - b_t) \cdot v_t^{\text{forward}}, & \text{if } \Delta(b_t, h_t) \leq 3 \\ 0, & \text{otherwise} \end{cases}$$

where $v_t^{\text{forward}}$ is the robot normalized forward vector in the world frame, and the vector $(h_t - b_t)$ is also normalized.

During training, the episode terminates if there is a collision between the humanoid and the robot. The robot receives a bonus reward of $+10$ if the robot successfully maintains a safety distance between $1m$ and $2m$ to the humanoid and points to the humanoid for at least 400 simulation steps. The criteria for 'facing the humanoid' is computed by the dot product of the robot's forward vector and the vector pointing from the robot to the humanoid, with the threshold of $> 0.5$. We assume the robot has access to an arm depth camera ($224 \times 171$ with the horizontal field of view (hFOV) of $55$), an arm RGB camera ($480 \times 640$ with hFOV of $47$), a binary human detector (1-dim), and the relative pose of the humanoid in polar coordinate system (2-dim). In addition, a slack reward of $-0.1$ is given to encourage the agent to find the humanoid as soon as possible. In all episodes, to make sure that the robot learns to find the humanoid, the robot location is initialized at least $3m$ away from the humanoid. The final social navigation reward is as follows:

$$r_t^{\text{social-nav}} = 10 \mathbb{1}^{\text{success}} + r_t^{\text{distance}} + 3 r_t^{\text{orientation}} - 0.1.$$

During evaluation, the total episode length is 1500 steps and the episode terminates if there is a collision between the humanoid and the robot. In addition, the robot location is initialized at least $5m$ away from the humanoid, and the initial locations of the robot and the humanoid, and the humanoid

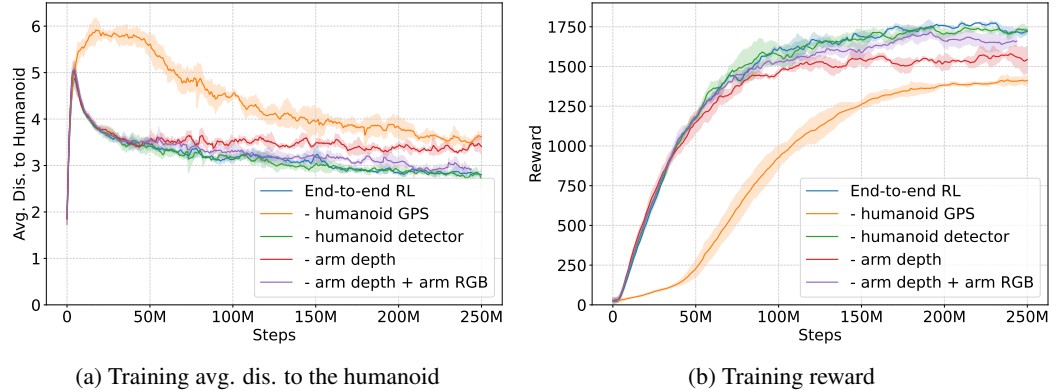

(a) Training avg. dis. to the humanoid          (b) Training reward

Figure 5: **Social Navigation training curves.** We plot the training average distance to the humanoid and reward for the social navigation baselines and ablations. We use 3 seeds for each model.

path are fixed across the baselines. This ensures that we have a fair comparison across different baselines.

For the social navigation task, the output space of the policy is the linear and angular velocities with the range of $-1$ and $+1$ (2-dim), followed by scaling it to $-10$ and $+10$, which is equivalent to $2.5m/s$ in the real world. We use the same maximum linear and angular velocities of $2.5m/s$ (rad/$s$) for both humanoid and robot. Since the Spot robot has a long body shape and cannot be represented by a single cylinder, we use a 2-cylinder representation, placed in the center and the front of the robot for collision detection. This ensures that the robot arm camera does not penetrate walls or obstacles while allowing the robot to navigate in a cluttered scene.

Fig. 5 shows the average distance between the humanoid and the robot and reward learning curve over the number of simulation steps for the end-to-end RL policy and its ablations. We see that the agent is able to improve the reward while minimizing the distance to the humanoid for finding and following the humanoid over training.

**Oracle for the Minimum Steps.** To compute the Finding Success Weighted by Path Steps and Following rate, we need to measure the optimal finding time $l$. This measure is similar to the optimal path length in navigation tasks, but in this case the target to navigate to is dynamic. We thus define the optimal path length $l$ as the minimum time that it would take for an agent to reach the humanoid if it knew the humanoid's trajectory in advance. Formally, let $h_i$ be the humanoid position at step $i$ and $r_i$ the minimum number of steps to go from the robot starting position to $h_i$, we define $l$ as:

$$l = \arg\min_i(r_i < i), \tag{1}$$

measuring the earliest time where the robot will be able to find the humanoid. To compute this measure, we split the humanoid trajectory into equally spaced waypoints, we then use a path planner to measure the number of steps it would take to reach each waypoint from the robot starting position and take the earliest waypoint satisying Eq. 1. Given this measure, the optimal following time corresponds to that of a robot which can find the humanoid in $l$ and follow it until the end of the episode, i.e. $E - l$, with $E$ being the episode length.

### A.2 SOCIAL REARRANGEMENT

We train all the rearrangement baselines using DD-PPO (Wijmans et al., 2019), distributing training across 4 NVIDIA A100 GPUs. Each GPU runs 24 parallel environments, and collects 128 steps for each update. We train with Adam (Kingma & Ba, 2014) using a learning rate of $2.5e^{-4}$. We use 2 PPO minibatches and 1 epoch per update, an entropy loss of $1e^{-4}$, and clip the gradient norm to 0.2. All policies are trained for 100M environment steps. The policy uses a ResNet-18 (He et al., 2016) visual encoder to embed the $256 \times 256$ depth input image into a 512 dimension embedding. The visual embedding is then concatenated with the state sensor values and passed through a 2-layer LSTM network with hidden dimension 512. The LSTM output is then set to an action and value prediction network. All methods use the same reward function specified as $+10$ for succeeding in the

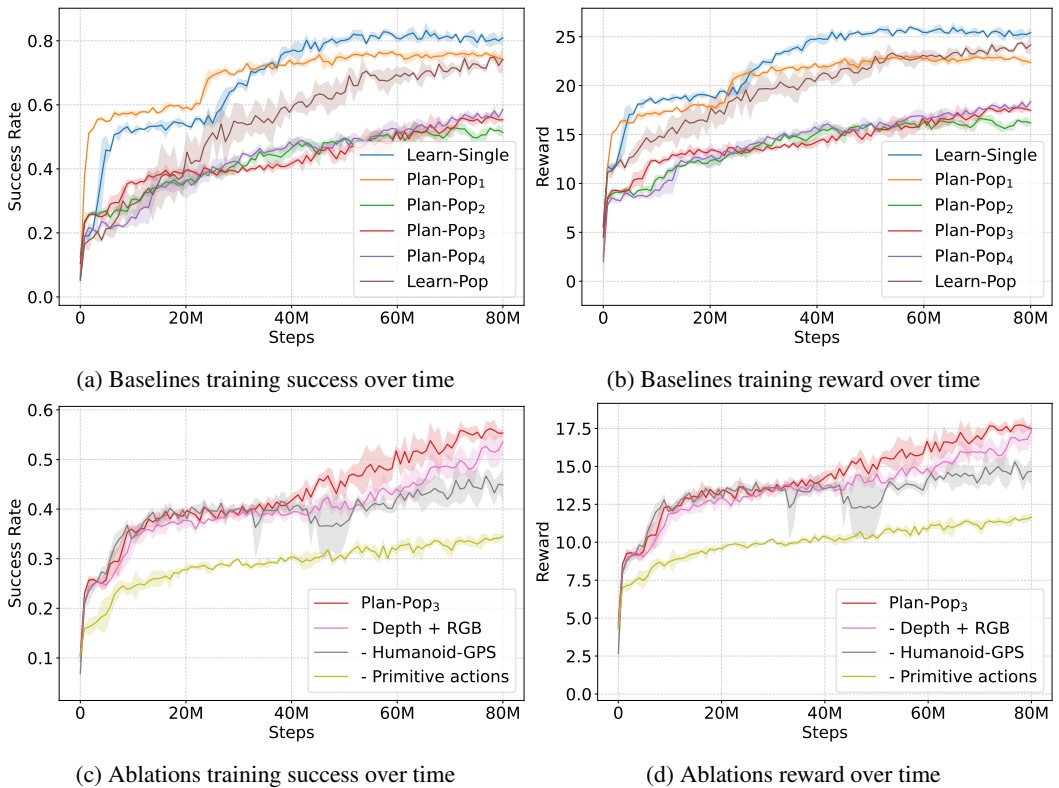

(a) Baselines training success over time      (b) Baselines training reward over time

(c) Ablations training success over time      (d) Ablations reward over time

Figure 6: **Social Rearrangement training curves.** We plot the training success and reward for the social rearrangement baselines **(top)** and ablations **(bottom)**. We use 3 seeds for each model.

overall task, $+5$ for completing any subgoal consisting of picking one of the target objects or placing an object at its goal, $-0.005$ penalty per simulator timestep to encourage faster completion, and a $-5$ penalty and episode termination if the agents collide. All baselines are trained with three different random seeds, and results are reported averaged across those seeds.

The final social rearrangement reward is as follows:

$$r_t^{\text{social-rearrange}} = 10 \cdot \mathbb{1}^{\text{success}} + 5 \cdot \mathbb{1}^{\text{subgoal}} - 5 \cdot \mathbb{1}^{\text{collision}} - 0.005.$$

Fig. 6 shows learning curves for all baselines and ablations on the social rearrangement task. We present the overall task success as well as the training reward for all approaches, averaged over 3 seeds. We observe that some baselines like Learn-Single and Plan-pop$_1$ are able to learn the task much faster than other baselines like Plan-pop$_{2,3,4}$ due to a simpler training setting. Among the ablations, removing the sensors used in original training make learning slower, with primitive actions having the most effect.

**Learned and Oracle skills.** For ease of learning, we adopt a two-layer policy architecture for all baselines, where a learned high-level policy selects a low-level skill to execute based on observations. The action space of the learned high-level policy consists of discrete selections from all possible combinations of skills and objects/receptacles allowed at each step. We work with a known, fixed library of low-level skills that can accomplish instructions like "navigate to the fridge" or "pick an apple." For the robot, we consider both learned skills and oracle skills that use privileged information from the environment. Additionally, we provide 4 'primitive' actions to the high-level policy that move the robot forward/backward or turn it left/right by a fixed amount. During training, we use the oracle low-level skills to train the high-level policies due to faster training speed, but present evaluation results with both oracle and learned skills. For the humanoid, low-level skills are always oracle, as described in Section 3.1.

The oracle navigation skill has access to a map of the environment, and plans a shortest path to the agent's destination. Manipulation oracle skills like "pick or place an apple" instantaneously attach the apple to the gripper, or place it at the target location. If the high-level policy chooses to execute an infeasible low-level action, such as attempting to pick an object that is out of reach (based on predefined pre-conditions), the action results in a no-op with no changes to the environment. If the robot approaches the humanoid within a short distance ($< 1.5m$) while executing a skill, the current skill is aborted, and the high-level policy replans its next action. This results in reactive behaviors, like the high-level policy commanding the robot to move backwards to give way to the humanoid in narrow corridors, or choosing to pick the second object after realizing that the humanoid is picking the first.

When using learned skills, we use the same 2-layer policy architecture, except use learned navigation, and learned pick/place skills, which operate entirely using robot depth and onboard sensors. These skills do not use privileged information, and hence are more prone to failures in the diverse set of scenes considered in our tasks. Refer to Section B for more detail on training and design of low-level skills.

When evaluating social rearrangement with learned low-level skills, we keep the high-level policy frozen, after training it with oracle skills. Hence the high-level policy is not robust to low-level execution failures. As a result we observe a considerable drop in the overall performance, when using learned skills (Table 3). This performance can potentially be improved by training the high-level policy with learned low-level skills in-the-loop, or by fine-tuning in this setting. We leave this to future work.

## B  Low-level Skill Training

We include the implementation details for training the low-level skills: navigation, pick, and place skills, which are coordinated by the high level policy described in Section 4.2.

**Navigation Skill.** Given the location of the target object, the robot uses neural network policies[4] to find the object, similar to PointNav (Anderson et al., 2018). This navigation skill is different from the social navigation policy since the former does not require the robot to continuously follow the humanoid while avoiding collision. The robot has access to an arm depth camera ($224 \times 171$ with hFOV of 55), and the relative pose of the target object in polar coordinate system (2-dim). Similar to the policy used in the social navigation task, the output space of the navigation skill is the linear and angular velocities with the range of $-1$ and $+1$ (2-dim), followed by scaling to $-10$ and $+10$. We also use the same 2-cylinder collision shape representation as the one in the social navigation task to ensure that the robot arm camera does not penetrate walls or obstacles while allowing the robot to navigate in a cluttered scene.

During training, the object navigation reward that encourages moving forward the target object $r_t^{\text{distance}}$ at time $t$ is defined as $\Delta(b_{t-1}, e_{t-1}) - \Delta(b_t, e_t)$, where $\Delta(b_t, e_t)$ is the shortest path distance between the robot location $b_t$ and the object $e_t$ at time $t$. To encourage the robot to orient itself toward the target object for improving grasping, it receives an additional 'orientation' reward $r_t^{\text{orientation}}$ when $\Delta(b_{t-1}, e_{t-1}) - \Delta(b_t, e_t) \leq 3$, in which the orientation reward penalizes the dot product of the robot forward vector and the robot to target object vector weighted by the scale of $5 \times 10^{-2}$. A navigation success reward of $+10$ is given if (1) the distance between the agent and the target object is less than $1.5m$, and (2) the dot product of the robot forward vector and the robot to target object vector $> 0.5$. To reduce the collision between the robot and the scene, a penalty of $-5 \times 10^{-3}$ is given if there is a collision. Finally, a slack reward of $-1 \times 10^{-2}$ is given to encourage the robot to find the target object as soon as possible. The final navigation reward is as follows:

$$r_t^{\text{nav}} = 10 \mathbb{1}^{\text{success}} + r_t^{\text{distance}} + 0.05 r_t^{\text{orientation}} - 0.005 \mathbb{1}_t^{\text{collision}} - 0.01.$$

During training, the episode terminates if the robot finds the object or reaches the maximum episode simulation step of 1500. In addition, the episode also terminates if the robot collides with a humanoid that walks randomly, similar to the setup in training social navigation policies. To make sure that the robot learns to navigate in complex scenes, the robot is placed at least $4m$ away from the target object

---

[4]In this section, we use 'policy' and 'skill' interchangeably to refer to a controller parameterized by neural networks. In addition, we use 'robot' and 'agent' interchangeably to refer to a reinforcement learning agent.

location. We train the navigation skill using DD-PPO, distributing training across 4 NVIDIA A100 GPUs, and with learning rate of $1 \times 10^{-4}$ and the maximum gradient norm of $0.2$. Each GPU runs 24 parallel environments, and collects 128 steps for each update. We use a long short-term memory networks (LSTM) policy with ResNet-18 (He et al., 2016) as the visual backbone and two recurrent layers, resulting nearly 8517k parameters. It takes about 300 million simulation steps (roughly 6 days of training) to reach $90\%$ navigation success rate using the above hardware setup.

**Pick Skill.** Given the location of the target object, the robot uses neural network policies to pick up an object by controlling the arm and moving the base (i.e., mobile pick, as defined in Gu et al. (2023)). The robot has access to (1) an arm depth camera ($224 \times 171$ with hFOV of 55), (2) the relative pose of the target object in a Cartesian coordinate system (3-dim), (3) the arm joint angles (7-dim), (4) a binary holding detector if the robot is holding an object (1-dim), and (5) the relative pose of arm end-effector to the target resting location in a Cartesian coordinate system (3-dim). The output space of the pick skill is (1) the linear and angular base velocities with the range of $-1$ and $+1$ (2-dim), followed by scaling to $-10$ and $+10$, (2) the delta arm joint angles applied to the arm with the range of $-1$ and $+1$ (7-dim), followed by a scaling factor of $5 \times 10^{-2}$, and a binary command to snap/desnap the object to the end-effector (1-dim). The robot can only pick up the object if the distance between the end-effector and the object is less than $0.15m$ (we teleport the object to the end-effector to simulate the grasping).

During training, before picking up the object, the pick reward that encourages the arm to move toward the object $r_t^{\text{move}}$ at time $t$ is defined as $\Delta(c_{t-1}, e_{t-1}) - \Delta(c_t, e_t)$, where $\Delta(c_t, e_t)$ is the geodesic distance between the robot's arm end-effector location $c_t$ and the object $e_t$ at time $t$. After picking up the object, the retract-arm reward that encourages the robot to retract the arm $r_t^{\text{retract}}$ at time $t$ is defined as $\Delta(c_{t-1}, q_{t-1}) - \Delta(c_t, q_t)$, where $\Delta(c_t, q_t)$ is the distance between the robot's arm end-effector location $c_t$ and the target end-effector resting location $q_t$ at time $t$. The robot receives the success reward of $+2$ if the robot (1) picks up the right target object, and (2) $\Delta(c_t, q_t)$ is less than $0.15m$. Finally, a slack reward of $-5 \times 10^3$ is given to encourage the robot to pick up the target object as soon as possible. The final pick reward is as follows:

$$r_t^{\text{pick}} = 2 \mathbb{1}^{\text{success}} + r_t^{\text{move}} + r_t^{\text{retract}} - 0.005.$$

During training, the episode terminates if the robot (1) picks up the wrong object or drops the object, both with the penalty of $-0.5$, (2) reaches maximum simulation steps of 1250, or (3) successfully picks up the right target object and retracts the arm, with the success reward of $+2$. To make sure that the robot learns to orient itself to pick up the object, the robot is placed at least $3m$ away from the target object location. We train the pick skill using DD-PPO, distributing training across 8 NVIDIA GPUs, and with learning rate of $3 \times 10^{-4}$ and the maximum gradient norm of $0.2$. Each GPU runs 18 parallel environments, and collects 128 steps for each update. We use a long short-term memory network (LSTM) policy with ResNet-18 as the visual backbone and two recurrent layers, resulting nearly 8540k parameters. It takes about 100 million simulation steps (roughly one day of training) to reach $90\%$ pick success rate using the above hardware setup.

**Place Skill.** Given the location of the goal location, the robot uses neural network policies to place an object by controlling the arm and moving the base (i.e., mobile place Gu et al. (2023)). For consistency, the input space of the place skill has the exact same input space as the pick skill. The output space of the place skill also shares the same output space of the pick skill. The robot can only place the object if the distance between the end-effector and the target place location is less than $0.15m$ (we teleport the object from the end-effector to the target place location).

During training, before placing the object, the place reward that encourages the robot to move close to the target location $r_t^{\text{move}}$ at time $t$ is defined as $\Delta(c_{t-1}, e_{t-1}) - \Delta(c_t, e_t)$, where $\Delta(c_t, e_t)$ is the distance between the robot's arm end-effector location $c_t$ and the object $e_t$ at time $t$. After placing the object, the retract-arm reward $r_t^{\text{retract}}$ is the same as the one in pick skill to learn to reset the arm. In addition, the robot receives an addition bonus reward $r_t^{\text{bonus}}$ of $+5$ if the robot places the object in the right location. Finally, the robot receives the success reward of $+10$ if (1) the robot places the object in the right location, and (2) $\Delta(c_t, q_t)$ is less than $0.15m$. A slack reward of $-5 \times 10^3$ is given to encourage the robot to place the object as soon as possible. The final place reward is as follows:

$$r_t^{\text{place}} = 10 \mathbb{1}^{\text{success}} + r_t^{\text{bonus}} + r_t^{\text{move}} + r_t^{\text{retract}} - 0.005.$$

| | S↑ | SPS↑ | F↑ | CR↓ | BYR | TD↓ | FD↓ |
|---|---|---|---|---|---|---|---|
| Heuristic Expert | 1.00 | 0.97 | 0.51 | 0.52 | 0.24 | 2.56 | 1.72 |
| End-to-end RL | $0.97_{\pm0.00}$ | $0.65_{\pm0.00}$ | $0.44_{\pm0.01}$ | $0.51_{\pm0.03}$ | $0.19_{\pm0.02}$ | $3.43_{\pm0.07}$ | $1.70_{\pm0.04}$ |
| - humanoid GPS | $0.76_{\pm0.02}$ | $0.34_{\pm0.01}$ | $0.29_{\pm0.01}$ | $0.48_{\pm0.03}$ | $0.13_{\pm0.00}$ | $5.18_{\pm0.11}$ | $1.64_{\pm0.02}$ |
| - humanoid detector | $0.98_{\pm0.00}$ | $0.68_{\pm0.00}$ | $0.37_{\pm0.01}$ | $0.64_{\pm0.05}$ | $0.16_{\pm0.03}$ | $3.44_{\pm0.03}$ | $1.67_{\pm0.09}$ |
| - arm depth | $0.94_{\pm0.01}$ | $0.54_{\pm0.01}$ | $0.19_{\pm0.01}$ | $0.71_{\pm0.08}$ | $0.15_{\pm0.01}$ | $4.94_{\pm0.03}$ | $1.91_{\pm0.27}$ |
| - arm depth + arm RGB | $0.96_{\pm0.00}$ | $0.61_{\pm0.01}$ | $0.38_{\pm0.02}$ | $0.55_{\pm0.04}$ | $0.17_{\pm0.02}$ | $3.74_{\pm0.05}$ | $1.82_{\pm0.05}$ |

Table 2: **Social Navigation baseline results.** We report three additional metrics: (1) *Backup-Yield Rate (BYR)*, (2) *The Total Distance between the robot and the humanoid (TD)*, and (3) *The 'Following' Distance between the robot and the humanoid after the first encounter (FD)*.

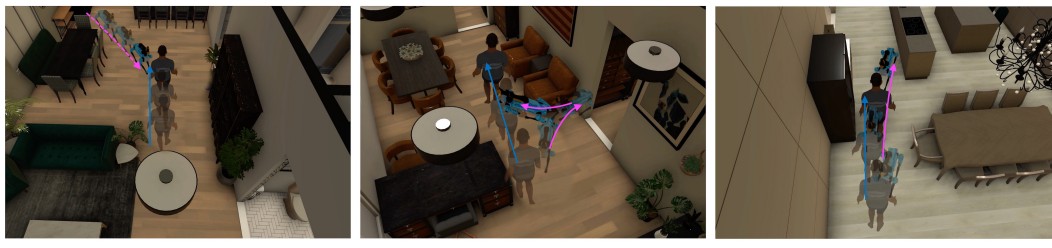

— Humanoid path ⋯ Find Phase — Follow Phase

Figure 7: **Illustration of Social Navigation end-to-end RL policy's behavior.** The robot finds the humanoid (**left**). The robot yields to the humanoid by doing a 'three-point-turn' motion (**center**). The robot yields to the humanoid by backing up (**right**).

During training, the episode terminates if the robot (1) places the object in the wrong location, (2) reaches maximum simulation steps of 1250, or (3) successfully places the right target object and retracts the arm, with the success reward of $+10$. To make sure that the robot learns to orient itself to place the object, the robot is placed at least $3m$ away from the target object location. Similar to the one in pick skill, we train the navigation skill using DD-PPO, distributing training across 8 NVIDIA GPUs, and with learning rate of $3 \times 10^{-4}$ and the maximum gradient norm of $0.2$. Each GPU runs 18 parallel environments, and collects 128 steps for each update. We use a long short-term memory network (LSTM) policy with ResNet-18 as the visual backbone and two recurrent layers, resulting nearly 8540k parameters. It takes about 50 million simulation steps (roughly half of a day of training) to reach $90\%$ place success rate using the above hardware setup.

## C    DETAILED COMPARISON RESULTS

Here we provide additional detailed results comparing the performance of different baselines along additional metrics.

### C.1    SOCIAL NAVIGATION

In this section, we provide a detailed analysis of the Social Navigation task, studying the behavior of the different baselines at evaluation time.

#### C.1.1    ADDITIONAL METRICS

In Table 2, we further report three additional metrics: (1) *Backup-Yield Rate (BYR)*: How often did the robot do a backup or yield motion to avoid collision when the human is nearby? We define a 'backup motion' as a backward movement by the robot to avoid collision with the humanoid when the distance between them is less than 1.5 meters. Furthermore, we define a 'yield motion' as a robot's motion aimed at avoiding collision with the humanoid when the distance between them is less than 1.5 meters, and the robot's velocity is less than $0.1m/s$; (2) *The Total Distance between the robot and the humanoid (TD)*: What was the L2 distance (in meter) between the robot and the humanoid over the total number of episode steps?, and (3) *The 'Following' Distance between the robot and the humanoid after the first encounter* (FD): What was the L2 distance between the robot

and the humanoid after the robot finds the humanoid? In summary, the backup-yield rate lets us quantitatively measure the frequency of backup and yield motions, and the two distance matrices provide an observation of how close the robot and the humanoid are during the finding and following stages. Ideally, the FD should be between 1-$2m$, while policies with higher SPS have lower TD.

### C.1.2 Additional Analysis

In this section, we provide additional analysis for the end-to-end RL policy and its ablations. Fig. 7 provides an example of how the robot moves to find the humanoid and produces a backup motion to yield to the humanoid by anticipating where the humanoid will walk next. For the ablation baseline without the humanoid GPS, we observe that it learns two types of finding humanoid strategies. The first strategy is that the robot randomly walks to increase the chances of finding the humanoid. The second strategy is that the robot keeps rotating until there is a humanoid in sight (i.e., scanning the environment), captured by the humanoid detector. As a result, compared to the method with the humanoid GPS, the one without the humanoid GPS needs more steps to find the humanoid (lower SPS). It also tends to lose track of the humanoid when the humanoid walks into another room, leading to the case that the humanoid is not in sight, and the robot needs to find the humanoid again (lower following rate).

For the ablation baseline without the humanoid detector, we observe that it has worse following performance (7% drop), and a higher collision rate (13% increase) than those of the one with the humanoid detector. This occurs because while the humanoid GPS offers a relative L2 distance to the humanoid, it does not account for scenarios where a wall or obstacle obstructs the robot's view. As a result, the robot has no idea if there is a wall with a low L2 distance to the humanoid, leading to a lower following rate and higher collision rate. Providing the humanoid detector allows the agent to 'see' if there is a humanoid there, and thus follow the humanoid.

For the ablation baseline without the arm depth, we find that it has the highest collision rate (leading to the lowest following rate). This is because the arm depth provides useful information to know and record where the empty space/obstacles are (incorporated with the LSTM memory) when the robot needs to avoid collision. As a result, we find that the baselines with the arm depth or arm RGB tend to have a lower collision rate.

Finally, for the ablation baseline with the arm depth being replaced by the arm RGB, we find that it has a slightly higher collision rate than the one with the arm depth. This is because the arm depth provides information about the distance to obstacles, leading to a better moving strategy to avoid collisions.

Overall, we find that the end-to-end RL policy and its ablations have a comparable Backup-Yield Rate, suggesting that humanoid avoidance motion is learned through RL training. At the same time, the RL policy and its ablations can maintain a close distance to the humanoid (low Following Distance). But still, there is a performance gap between the RL policy and the Heuristic Expert in terms of SPS and the following rate. This leaves room for future improvement.

## C.2 Social Rearrangement

Here we present additional metrics and analysis for social rearrangement. We also describe the different ablations in more details.

### C.2.1 Additional Metrics

**Collision Rate (CR)**. Together with the success rate and relative efficiency, we are interested in measuring whether the social rearrangement agents can complete tasks safely, without colliding with the humanoid agent. Thus, we measure the collision rate (CR) in both the train-population and the zsc-population settings. Following Sec. 3.2, we define CR as the proportion of episodes containing collisions between the robot and the humanoid. We report the results in Table 3, along with the Success rate (SR) and Relative Efficiency (RE) metrics. We observe similar trends to the success rate measure, with Learn-Single, Plan-Pop$_1$, having low CR with the training population but high CR with ZSC population ($0.09 \rightarrow 0.23$ with train-pop for Plan-pop$_1$). Plan-Pop$_4$ obtains the best collision rate in the training population, with a rate of 9%, despite having a more diverse population than other baselines. This is because one of the agents in the training population for Plan-pop$_4$ stays

| Method | Train-pop-eval | | | | ZSC-pop-eval | | | |
|---|---|---|---|---|---|---|---|---|
| | SR↑ | RE↑ | CR↓ | RC↑ | SR↑ | RE↑ | CR↓ | RC↑ |
| Learn-Single | $\mathbf{98.50}_{\pm\mathbf{0.48}}$ | $\mathbf{159.2}_{\pm\mathbf{1.0}}$ | $0.12_{\pm0.12}$ | $0.49_{\pm0.00}$ | $50.94_{\pm39.55}$ | $106.02_{\pm34.32}$ | $0.25_{\pm0.33}$ | $0.45_{\pm0.02}$ |
| Plan-Pop$_1$ | $91.2_{\pm2.63}$ | $152.4_{\pm5.4}$ | $\mathbf{0.09}_{\pm\mathbf{0.09}}$ | $0.46_{\pm0.00}$ | $50.44_{\pm39.02}$ | $109.75_{\pm34.63}$ | $0.23_{\pm0.31}$ | $0.43_{\pm0.05}$ |
| Plan-Pop$_2$ | $66.89_{\pm1.47}$ | $110.06_{\pm6.83}$ | $0.10_{\pm0.10}$ | $0.50_{\pm0.00}$ | $70.23_{\pm7.02}$ | $102.13_{\pm11.10}$ | $\mathbf{0.15}_{\pm\mathbf{0.17}}$ | $0.52_{\pm0.05}$ |
| Plan-Pop$_3$ | $77.79_{\pm2.86}$ | $118.95_{\pm6.04}$ | $0.12_{\pm0.12}$ | $0.48_{\pm0.01}$ | $71.79_{\pm7.38}$ | $101.99_{\pm15.18}$ | $0.17_{\pm0.19}$ | $0.53_{\pm0.05}$ |
| Plan-Pop$_4$ | $72.42_{\pm1.32}$ | $105.49_{\pm1.7}$ | $\mathbf{0.09}_{\pm\mathbf{0.09}}$ | $\mathbf{0.55}_{\pm\mathbf{0.00}}$ | $71.32_{\pm6.47}$ | $103.53_{\pm9.8}$ | $0.16_{\pm0.17}$ | $0.53_{\pm0.04}$ |
| Learn-Pop | $92.20_{\pm2.21}$ | $135.32_{\pm3.43}$ | $0.15_{\pm0.15}$ | $0.50_{\pm0.00}$ | $48.52_{\pm35.51}$ | $99.80_{\pm31.02}$ | $0.26_{\pm0.33}$ | $0.46_{\pm0.02}$ |
| + learned skills | $41.09_{\pm2.15}$ | $79.63_{\pm1.76}$ | $0.37_{\pm0.19}$ | $0.12_{\pm0.12}$ | $21.44_{\pm18.26}$ | $76.45_{\pm9.23}$ | $0.17_{\pm0.17}$ | $0.12_{\pm0.12}$ |
| - depth + RGB | $76.70_{\pm3.15}$ | $110.04_{\pm3.05}$ | $0.13_{\pm0.14}$ | $0.49_{\pm0.02}$ | $70.89_{\pm8.18}$ | $100.16_{\pm14.79}$ | $0.16_{\pm0.18}$ | $0.54_{\pm0.04}$ |
| - Humanoid-GPS | $76.45_{\pm1.85}$ | $108.96_{\pm2.66}$ | $0.18_{\pm0.18}$ | $0.49_{\pm0.01}$ | $68.70_{\pm6.75}$ | $98.58_{\pm10.32}$ | $0.22_{\pm0.24}$ | $0.53_{\pm0.05}$ |
| - Primitive actions | $85.71_{\pm1.58}$ | $124.36_{\pm3.79}$ | $0.32_{\pm0.32}$ | $\mathbf{0.55}_{\pm\mathbf{0.00}}$ | $\mathbf{76.80}_{\pm9.66}$ | $\mathbf{111.97}_{\pm\mathbf{10.91}}$ | $0.33_{\pm0.34}$ | $\mathbf{0.58}_{\pm\mathbf{0.04}}$ |

Table 3: **Social Rearrangement baseline results.**

in place does no part of the task, thus reducing the total number of collisions. In the ZSC setting, Plan-Pop$_2$ obtains the lowest collision rate, though the rates are comparable across Plan-Pop$_{2,4}$.

**Ratio of Completion (RC)**. We also report, for all our baselines, the ratio of the task completed by the robot, which measures the proportion of objects that were rearranged by the robot agent. A value of 1.0 indicates that the task is completely done by the robot, whereas 0.0 indicates that the task was done by the humanoid alone. Values closer to 0.5 indicate that the task was split among the robot and humanoid, resulting in increased efficiency. We show the results in Table 3. Almost all baselines achieve a RC close to 0.5 with training population. Learn-Single achieves a RC close to 0.5 in the train population, showing that both agents learned to split the task to perform it efficiently. Plan-Pop$_1$ shows a slight decrease in RC, which is consistent with the drop in the SR, indicating that agents are still splitting the task evenly, while overall achieving lower success rate. Plan-pop$_4$ has the highest train-pop RC, because one of its training population agents complete no part of the task, requiring the robot to rearrange both objects. The results on ZSC population follow success rates, with Plan-Pop$_{2,3,4}$ achieving higher RC, since the agents are trained to rearrange either object. In comparison, Learn-Single, Plan-pop$_1$ and Learn-Pop have slightly reduced RC due to inability to generalize to partners that rearrange different objects than their training population.

### C.2.2 Additional Ablation Results and Analysis

In the main paper we presented ablation experiments where we: (1) replaced oracle skills with learned skills, (2) replaced depth arm camera with RGB, (3) Removed the humanoid GPS. Here, we present an additional ablation, where we remove some primitive actions like move backwards, forwards, turn left or right from the action space of the high-level policy. This measures the effect of these primitive navigation actions in the Social Rearrangement task, called - *Primitive actions*. The robot agent is trained in the same setting as Plan-Pop$_3$, but without the four low level navigation actions. We report the results in Table 3. As we can see, the collision rate (CR) increases significantly, showing that the primitive navigation actions are essential to reduce collisions between both agents. At the same time, removing the navigation actions improves the success rate and RE both in the Train-Pop and ZSC-pop settings. This is because the agent does not spend time making way for the humanoid, which allows to complete the task in a lower number of steps. Despite being faster at completion, the high CR makes this baseline non-ideal as it reduces the "safety" of human or humanoid collaborators.

### C.2.3 Zero-shot population details

Here we describe the ZSC population used in our experiments in detail, and their effect in the behavior of the trained agents in the zero-shot coordination setting.

The 10 ZSC population collaborators used in ZSC eval are created as follows: 3 are trained checkpoints from the training of Learn-Single, 3 are trained checkpoints from the training run of Learn-Pop and 4 are planner-based humanoids, where 1 picks up both objects, 2 pick up one of the two, and 1 stays still. For the learned checkpoints, we only use the learned policy for the humanoid, and discard the learned robot policy. As a result, most baselines have seen about 1/3 of the ZSC population during training, and need to generalize to 2/3 of the population. In general, the learned ZSC evaluation agents tend to focus on rearranging one of the 2 objects in the environment, while the planner-based agents follow their scripted behavior.

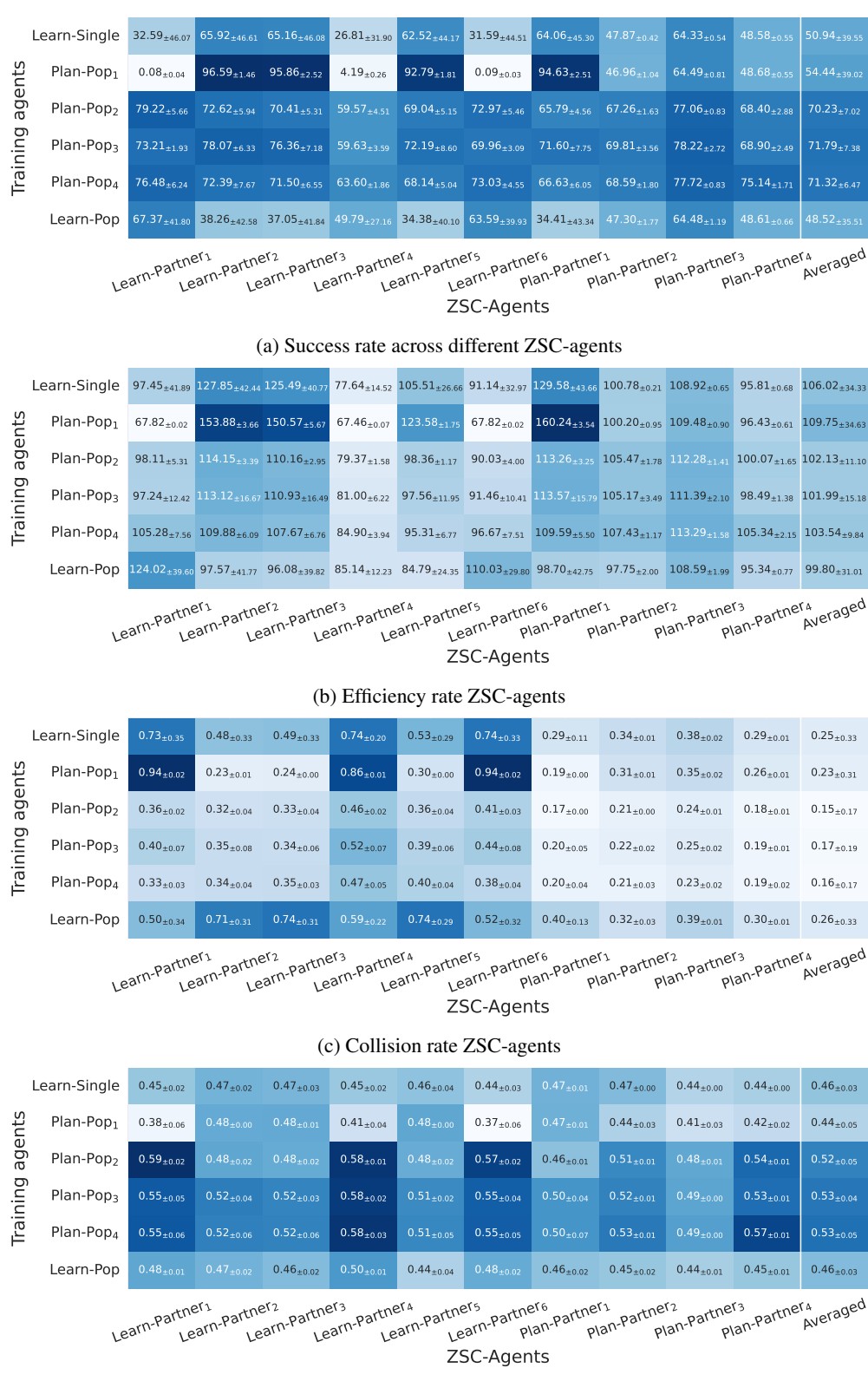

Figure 8: **Zero-shot coordination.** We report the performance of the baseline agents in the zero-shot coordination setting. Each row corresponds to one of the baselines and the columns represent the different types of zero-shot coordination agents.

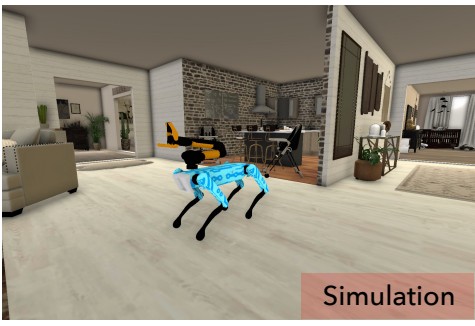 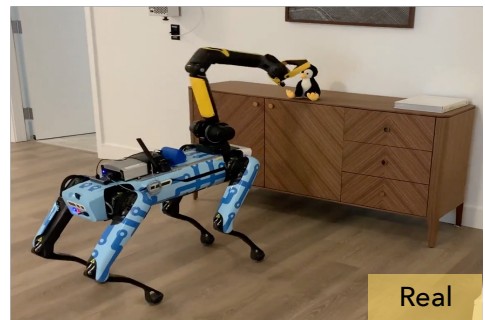

Figure 9: **Robot Embodiment.** Spot robot in the simulation environment is designed to minimize the embodiment gaps to the robot in the physical world.

We measure the trained agents' performance when evaluated with different agents from the ZSC-pop-eval population. In Figs. 8a, 8b, 8c and 8d, we show the success rate (a), efficiency rate (b), collision rate (c) and ratio of completion (d) of the different baseline trained agents under this setting. Each row corresponds to one of the baselines, trained and averaged across three seeds, and each column corresponds to one of the 10 ZSC evaluation agents. The last column corresponds to the ZSC-pop-val results in Fig. 4.

The Learn-Single agent overfits to one type of partner, learning to split the task by focusing on rearranging a single object and letting the other agent rearrange the remaining one. When evaluated with a ZSC partner, it will either be highly successful and efficient, if the evaluation partner was trained to pick the opposite object, or exhibit low performance if the evaluation agent is focused on the same object. For this reason the success rate in the first row of Fig. 8a is close to 33% and 66% for the Learn-Single agent, corresponding to one or two of the 3 training seeds matching the ZSC-partner.

Plan-Pop$_1$ exhibits a similar behavior to the Learn-Single baseline, but in this case, the agent is trained with a planner that always focuses on the same object, which makes Plan-Pop$_1$ focus on the opposite objects across all the training seeds. As a result, the agent has a success rate close to 100% or 0% for the different ZSC evaluation agents. As expected, it also exhibits a high success rate when partnering with the Plan$_1$ agent. Because of this, the agent also exhibits a significant increase in relative efficiency when partnering with certain agents. As a results it is the method with highest relative efficiency, when averaged across ZSC-agents, despite exhibiting high variance.

Plan-Pop$_2$ is trained with a population of agents arranging one of the two objects at random and therefore it cannot specialize anymore by always focusing on the same object. As a result, the agent shows much lower variance across the members of the ZSC-population, resulting in a higher success rate. At the same time, it needs to adapt to the behavior of the ZSC partnering agent, which results in lower peak efficiency than Plan-Pop$_1$ or Learn-Single, resulting on a lower relative efficiency on average. The average collision rate is also reduced relative to the previous baselines, with a lower collision rate with planning-based partners than the learning-based ones. Interestingly, this trend of lower CR with ZSC plan-agents holds across all baselines. We believe this is because ZSC plan-agents stop after finishing their portion of the task, while ZSC learned-agents continue to move in the environment (since our reward function does not penalize this). As a result, the chances of colliding with ZSC plan-partners is higher than with ZSC learned partners.

Plan-Pop$_{3,4}$ exhibit very similar performance across the different agents in the ZSC population. Their success rate shows a slight improvement with respect to Plan-Pop$_2$, whereas the relative efficiency decreases slightly for Plan-Pop$_3$ and increases slightly for Plan-Pop$_4$, though not significant. In general, adding an extra agents to Plan-Pop$_4$ that remains still does not seem to change the performance. To improve performance of learned coordination robot policies, we might need to incorporate other types of diversities, like humanoid speed, instead of just which object they rearrange.

### C.3 QUALITATIVE EXAMPLE

We also show in Fig. 10 an example episode of social rearrangement. We use Plan-Pop$_3$ as the baseline policy with learned low-level skills. In this example, the task is split amongst both agents, with each rearranging one of the goal objects. Frames 1,3 show the robot and humanoid picking up objects. 4,5 show them placing each object in its desired location. Frame 2 shows a scenario where the humanoid and robot cross paths, and the robot backs up to let the humanoid pass before continuing, avoiding a collision.

## D ROBOT

We use the Boston Dynamics (BD) Spot robot as the robot agent (Fig. 9) due to its robust hardware for real world deployment of trained policies in the future. Spot is a quadruped robot with five pairs of depth and RGB cameras (front-left, front-right, left, right, back), and a 7 degree of freedom arm with one pair of depth and RGB cameras mounted on the gripper. The arm depth camera has a $224 \times 171$ resolution and hfov of 55. Spot robots are capable of grasping rigid objects, climbing up/down stairs, and outdoor/indoor navigation given users' control input using BD's Spot control APIs. Its versatility and robustness make it an ideal mobile platform for studying sensing, manipulation, and human-robot interaction. On the simulation side, the Spot robot simulation moves its base by commanding linear and angular velocities (2D, continuous) and can move backwards if turning or moving forwards results in collision. The action space of the robot is continuous forward and angular velocity in the local frame of the robot (2-dim) for navigation and delta joint angles (7-dim) for the arm.

## E SCENES

We incorporate scene assets from the Habitat Synthetic Scenes Dataset (HSSD-200) (Khanna et al., 2023) in Habitat 3.0. HSSD is a dataset of 211 high-quality 3D homes (scenes) containing over 18k individual models of real-world objects. An example scene is shown in Fig. 11. For our experiments, we use a subset of 59 scenes and limit our objects to the YCB dataset (Calli et al., 2017) for simplicity. Specifically, we use 37 scenes from training, sampling 1000 episodes per scene, 12 for validation, with 100 episodes per scene and 10 for test, with 15 episodes in each scene. Note that the Habitat3.0 framework will be released with the complete HSSD dataset.

## F SIMULATOR DETAILS

### F.1 BENCHMARKING

We benchmark here Habitat 3.0 speed under varying scene sizes, different number of objects in the scene, and the type and number of agents. All tests are conducted on a single Nvidia V100 GPU. We report results on a single environment (Fig. 12, left) and 16 parallel Habitat 3.0 environments (Fig 12, right). For each measure we sample random actions for 300 steps per agent and report average and standard error results across 10 runs. All the solid blue curves correspond to small scenes sizes and two objects in the environment. For each agent, we always render a single depth image. Our small scene is $68.56m^2$ in size and has 1 bedroom and 1 bathroom, the medium scene is $136.11m^2$ in size with 3 bedrooms and 2 bathrooms, and the large scene is $846.15m^2$ in size with 4 bedrooms, 4 bothrooms, and a den and office space.

On a single environment, we obtain performances on the range of 140 to 250 FPS, depending on the setting. As we can see, varying the number of objects has no significant effect in the performance speed, while we notice significant differences when changing the scene size ($245\pm19$ FPS down to $154\pm14$). Switching from a single spot agent to two spot agents drops the performance from $245\pm19$ to $150\pm1$, having a similar effect to varying the scene size. The humanoid is also slower than the Spot robot ($245\pm19$ vs $155\pm26$), due to the higher number of joints in the skeletal model. However, the difference between robot and humanoid agents, becomes much less pronounced when switching to the two agent setting, obtaining comparable performances between the Robot-Robot and Human-Robot settings ($150\pm1$ vs. $136\pm1$). Since humanoid-robot simulation is the primary use-case of Habitat3.0, this is a positive signal, that shows that adding a humanoid over a robot does not decrease simulation

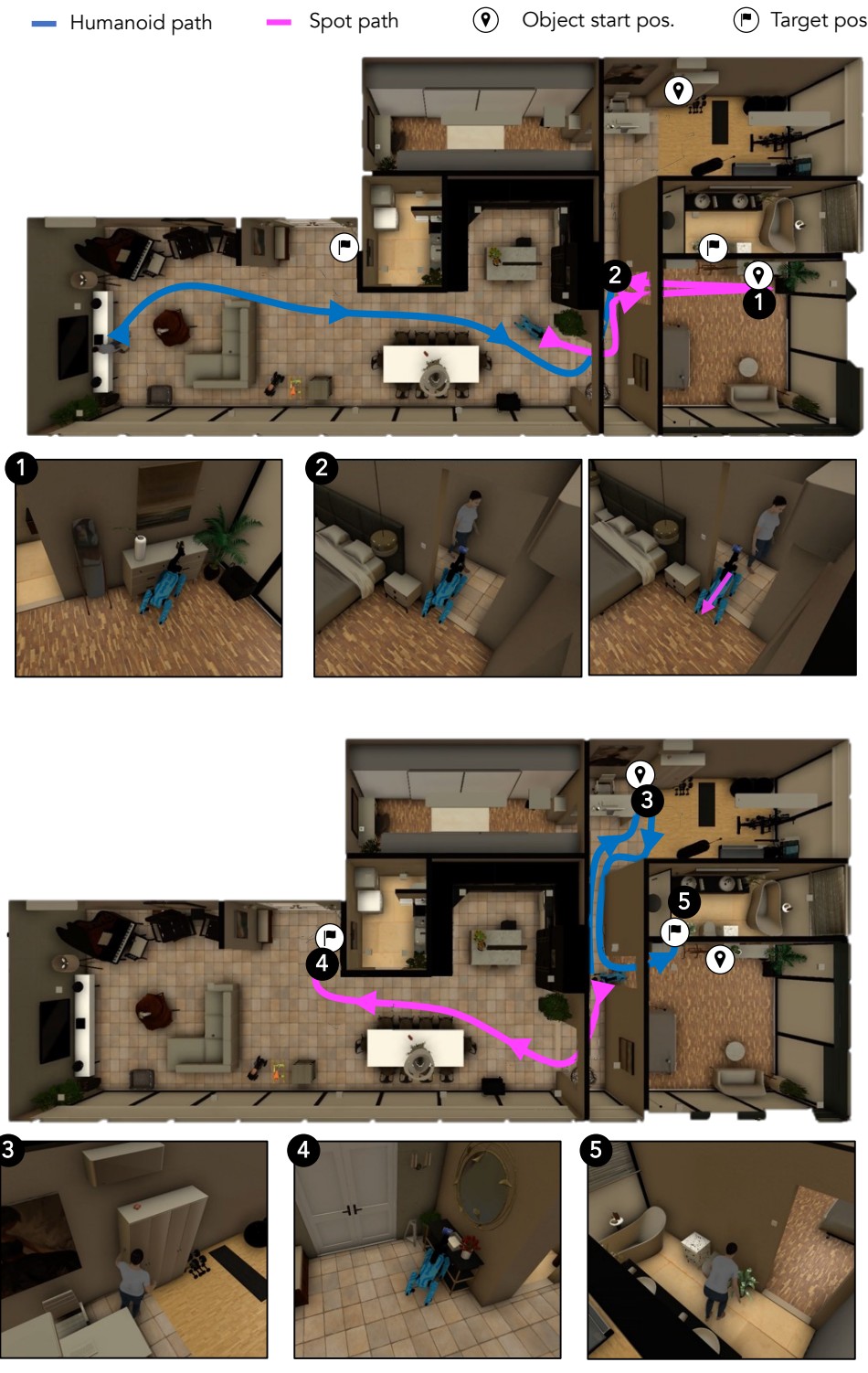

Figure 10: **Social Rearrangement Example**: We show an example behavior from the Plan-Pop$_3$ baseline with learned low-level skills. Here, the agents split the rearrangement task, with each picking one of the objects (frames 1, 3) and placing them in their target location (frames 4, 5). In frame 2, we observe an emergent collaborative behavior: the robot moves backward to let the humanoid pass through the hallway, before continuing with its task, avoiding a collision.

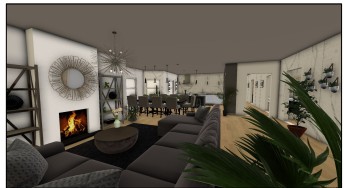 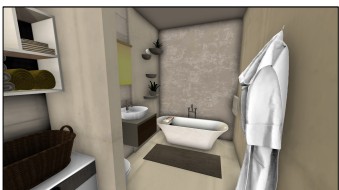 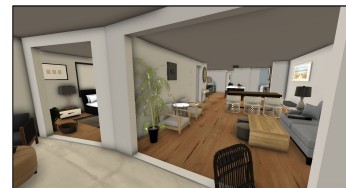

Figure 11: **Example Scene.** Habitat 3.0 is based on HSSD (Khanna et al., 2023) scenes. An example scene which includes various types of objects and scene clutter has been shown.

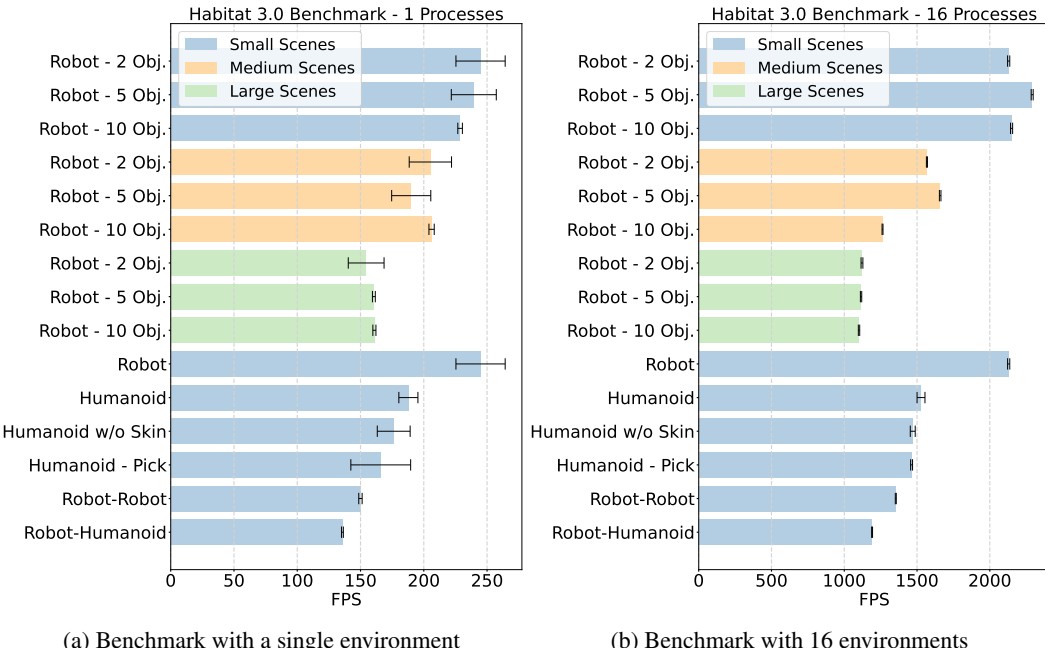

(a) Benchmark with a single environment

(b) Benchmark with 16 environments

Figure 12: **Benchmark results in Habitat 3.0**. We study the effect of varying scene size, number of objects, type of agents and single or multi-agent.

speed. We don't notice a difference in performance between representing the humanoid as a skeleton or applying linear blend skinning, implying that the visual realism of skinned humanoids has very little effect on our simulation speed. We also don't notice significant differences in performance between the picking or navigation actions.

We observe similar trends when switching to 16 environments, obtaining promising scaling results, with performances in the range of 1100 to 2290 FPS. Note that these 16 environments are still on a single GPU, and Habitat 3.0 natively supports environment parallelization. As a result, the most common simulation speed that users will experience when training policies is between 1100 to 2290 FPS depending on the scenario.

## F.2 COMPARISON WITH EXISTING SIMULATORS

Habitat 3.0 is designed to support efficient simulation of tasks with humans and robots in indoor environments. Since Habitat 3.0 is built upon the Habitat platform, it also supports features present in Habitat 1.0 and Habitat 2.0, including physics simulation, support for multiple types of robots, articulated objects or real scanned scenes. Our work opens an opportunity to update Embodied AI tasks previously studied in the Habitat platform with humanoid simulation or human in the loop interaction, allowing to train robots that can better interact with humans.

In Tab. 4 we provide a comparison of Habitat3.0 with some existing related simulators. Habitat3.0 provides support for both real robot models and humanoids, enabling to train policies for both types of

agents. Humanoids in Habitat3.0 are based on the SMPL-X model, as opposed to Authored Designs (**AD**). This enables to scale up the number of possible body models, and provide motions coming from motion captured data or motion generation models. The humanoids can also be controlled at different levels, including joints (**Jt**), inverse kinematics (**IK**) or high level commands, such as walk or pick up (**HL**). A key feature in our platform is the HITL interface, which allows to control the humanoid via a mouse-keyboard interface (⌨) or VR (🕶). Furthermore, we include a large number of authored multi-room scenes enabling training policies in a wide diversity of realistic indoor environments, and provides a significant increase in simulation speed, compared to other simulators with robot and humanoid support. Note that the speed is influenced by several factors, including rendering fidelity, physics simulation, resolution, hardware capabilities, and more (refer to Section F.1 for detailed benchmarking). As such, speed numbers between different simulators are not directly comparable and serve as rough reference points. We take the reported numbers in individual papers for single environment settings.

| | Robot | Humanoid | | | HITL | Type of | | #Authored | Speed |
|---|---|---|---|---|---|---|---|---|---|
| | Support | Support | Body Model | Control | Interface | Simulation | Multi-Agent | Scenes | (steps/s) |
| VirtualHome-Social | - | ✓ | AD | IK, HL | ⌨ | PP | ✓ | 50 | 10 |
| VRKitchen | - | ✓ | AD | Jt, IK | 🕶 | RBF, PP | - | 16 | 15 |
| SAPIEN | ✓ | - | - | - | - | RBF | - | - | 200-400 |
| AI2-THOR | ✓ | - | - | - | - | RBF, PP | ✓ | 120 | 90-180 |
| Habitat 2.0 | ✓ | - | - | - | - | RBF | - | 105 | 1400 |
| TDW | ✓ | ✓ | AD | IK, HL | 🕶 | RBF, PS | ✓ | 15 | $5-168$ |
| SEAN 2.0 | ✓ | ✓ | AD | HL | ⌨ 🕶 | RBF | ✓ | 3 | $3-60^{\dagger}$ |
| iGibson | ✓ | ✓ | AD | R | ⌨ 🕶 | RBF | ✓ | 50 | 100 |
| Habitat 3.0 | ✓ | ✓ | SMPL-X | Jt, IK, HL | ⌨ 🕶 | RBF | ✓ | 200 | 140-250 |

Table 4: **Comparison with related simulators.** We compare Habitat3.0 with other related simulators for Embodied AI supporting simulation of real robots or humanoids. Speeds were taken directly from respective publications or obtained via direct personal correspondence with the authors when not publicly available (indicated by †).
**Body Model:** Authored Design (AD), SMPL-X (Pavlakos et al., 2019).
**Control:** Joints (Jt), Inverse Kinematics (IK), High Level Commands (HL), Rigid Transforms (R).
**HITL Interface:** Mouse and Keyboard (⌨), Virtual Reality (🕶).
**Type of Simulation:** Rigid Body Physics (RBF), Pre-post-conditions (PP), Particle Simulation (PS).
**Speed:** The reported numbers correspond to benchmarks conducted by different teams using different hardware configurations to simulate diverse capabilities. Thus, these should be considered only as qualitative comparisons representing what a user should expect to experience on a single instance of the simulator (without parallelization).

## G  HUMAN-IN-THE-LOOP (HITL) EVALUATION

We test the ability of trained robotic agents to coordinate with real humans via our human-in-the-loop (HITL) infrastructure across 30 participants. Our user study consisted of 3 conditions: doing the task alone (*solo*), paired with a robot operating with *Learn-Single*, or paired with a *Plan-Pop$_3$* agent. We have a brief training step, where the users get accustomed to the tool, followed by solving each condition in a random order. However, as the users interact more with the tool, they get better at the overall task. We account for this learning effect by leveraging latin-square counter-balancing (Bradley, 1958) for the ordering of these conditions.

Each participant performs the task for 10 episodes per condition in one of the test scenes. We measure the collision rate (CR), task completion steps (TS) and Relative Efficiency (RE) across all episodes, and report them in Table 1. CR is the ratio of episodes that contain collisions with the robot. SR, RE are the same as defined in Sec 4.2. RE still captures the relative efficiency of task completion when the task is done with a robot and requires the computation of task completion steps with the robot relative to the human doing the task alone. However, unlike the automated evaluation (Train-pop-eval) and (ZSC-pop-eval), we cannot compute the RE per episode in HITL evaluation. Since humans exhibit learning effect when asked to do an episode multiple times, we cannot ask the human to do the same episode with and without the robot, preventing the computation of RE per episode. Instead, we fit a generalized linear mixed-effect model (GLMM) (Kaptein, 2016) with a Poisson distribution, for TS as the dependent variable and method/condition as the independent variable while controlling

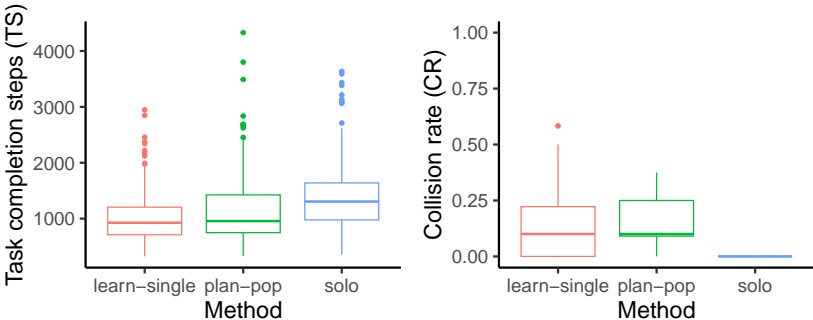

Figure 13: TS and CR across all participants in the user study.

| Method | Estimated mean difference in TS | p-value |
|---|---|---|
| Plan-pop - Learn-Single | 78.45 | 0.0533 |
| Solo - Learn-Single | 316.58 | <.0001 |
| Solo - Plan-pop | 238.12 | <.0001 |

Table 5: Post hoc pairwise comparison between the three conditions of the user study for TS.

for variations from participants and scenes using random intercepts (Bates et al., 2015). We then use the ratio between the estimated means of TS for Learn-Single and Plan-Pop conditions w.r.t the solo condition to compute their average RE respectively. For CR, we fit a logistic regression model with a binomial distribution, again controlling for the random effects of participants and scenes. We use lme4 package (Bates et al., 2015) in R version of 4.3.1 to fit the GLMMs. Table 1 shows the results over three conditions across the 30 users. Fig. 13 shows the TS over all successful episodes and CR over all episodes across the three conditions in HITL.

After fitting a GLMM model with Poisson distribution for TS, we also compute post hoc pairwise comparisons between the three conditions. We find that the difference in estimated mean of TS between Learn-Single and Plan-pop is not significant, while that between both these conditions and the solo condition is significant (Table 5). To the best of our knowledge, this is the first analysis of performance of learned collaboration agents when paired with real human partners in a realistic, everyday rearrangement task. The indication that the robot agents indeed make humans more efficient is very promising, with huge significance for the assistive-robotics community.

## H LIMITATIONS

In this section, we address the limitations of our work, focusing on our three primary contributions:
**Human simulation.** While our framework can accommodate various actions, our experiments utilize only walking and reaching behaviors. We implement the reaching behavior by obtaining static poses representing distinct reaching positions and interpolating between them to generate motion. For the walking motion, our approach for navigating across waypoints is suitable for path planners, but shows visual artifacts when the humanoid is rotating in place. While this approach suits the behaviors we consider, it may not be suitable for more complex motions, such as opening a cabinet or sitting down. This aspect will be explored in our future work. Additionally, we employ fixed linear blend skinning to represent the human, with static weight assignments for various poses. Consequently, certain pose configurations may exhibit skinning artifacts, as mentioned in the main paper.
**Human-in-the-loop tool.** There is a gap between the observations a human acquires through the HITL tool and those accessible to our humanoid in automated evaluations. Although we provide visual cues to bridge this gap (such as markers that indicate the direction from the human to the target objects), they still represent a different observation space between the two domains.
**Tasks.** Our current focus is on zero-shot coordination without communication, but introducing communication between the robot and human could potentially enhance task efficiency. Furthermore,

our models currently benefit from access to ground truth information regarding the initial and final object locations in the rearrangement task. A future challenge lies in integrating search capabilities into the rearrangement task. Additionally, we make the assumption that objects are initially placed on open receptacles. Handling rearrangements that involve searching for and interacting with articulated objects, like drawers, poses a greater challenge.

