# OpenReview forum: "Habitat 3.0: A Co-Habitat for Humans, Avatars, and Robots"
_ICLR.cc/2024/Conference — ICLR 2024 poster_

### Official Review · Reviewer_HoUA · 2023-10-31

**Soundness:** 4 excellent
**Presentation:** 4 excellent
**Contribution:** 4 excellent
**Rating:** 8
**Confidence:** 5

**Summary:**

This paper provides a simulator supporting humanoid avatars and robots for the study of collaborative human-robot tasks in home environments. Diverse and realistic human models are constructed, addressing the challenges of accurate modeling of the human body as well as the human-like appearance and motion diversity of the models. Besides, The platform supports interaction between a real person and a simulated robot via a mouse and keyboard inputs or VR interface, enabling human-in-the-loop simulation. This paper also investigates two collaborative human-robot interaction tasks, social navigation, and social reorganization, and provides insights into learned and heuristic baselines for both tasks in the simulator.

**Strengths:**

- This work presents a Co-Habitat for Humans, Avatars, and Robots, offering a simulation environment for humanoids and robots within a wide range of indoor settings, which can promote the development of human-robot tasks in the Embodied AI field.
- Habitat3 provides realistic human and robot simulation. In the process of realistic human modeling, this work addresses the challenges posed by efficiency realism and diversity in terms of appearance and movement.
- This work develops a Human-in-the-Loop evaluation platform within the simulator, allowing the control of humanoid robots using a mouse, keyboard, or VR devices. It provides a method for interacting and evaluating with real humans. Furthermore, it supports data collection and reproducibility during the interaction, offering a convenient tool for further research.
- This paper introduces two benchmark tasks for human-robot interaction, along with baselines for each task. This paper leverages end-to-end RL to study collaborative behaviors and examines the performance of various learning strategies. The Human-in-the-Loop evaluation in the social rearrangement task reveals potential avenues for improving social embodied agents.
- This simulator can be used for end-to-end reinforcement learning for robot agents, significantly reducing the time required for reinforcement learning. It also provides a validation environment for a broader range of robot agents, thus reducing potential risks to the environment and humans.

**Weaknesses:**

- Current robots are equipped with only a depth camera, human detector, and GPS, providing a relatively limited amount of information. It is worth considering whether additional sensor types, such as LiDAR and sound sensors, can be integrated in the future to enhance obstacle avoidance and navigation capabilities.
- It has been noted that in human simulation, fixed hand movements can lead to a decrease in visual accuracy. It may be considered to address the deformability of the skin during the simulation process and create hand motion animations during activities like grasping and walking.
- This simulator has a wide range of potential applications and can be further explored to implement other embedded artificial intelligence tasks, such as visual-language navigation.

**Questions:**

See Weaknesses

---

> ### Author Response · Authors · 2023-11-16
> **Response to reviewer HoUA**
>
> We thank reviewer *HoUA* for the insightful feedback. The questions and concerns are addressed below.
>
> **It is worth considering whether additional sensor types, such as LiDAR and sound sensors, can be integrated in the future to enhance obstacle avoidance and navigation capabilities.**
>
> Thank you for your suggestion. We agree that incorporating these sensors is useful, and it is on our roadmap. Since our work builds upon the Habitat platform, it already provides support for some of these sensors (for example, sound in [1]) and it would not be difficult to integrate them into future  tasks.
>
> **Fixed hand movements can lead to a decrease in visual accuracy. It may be considered to address the deformability of the skin during the simulation process and create hand motion animations during activities like grasping and walking.**
>
> Thank you for your comment. While we did not model hand movements in our work, we use the SMPL-X body format to represent the humanoids, which allows us to specify hand and finger positions. More realistic hand movement could be addressed by recording hand keypoints in motion capture data or by using generative models for hand poses [2].
>
>
> **This simulator has a wide range of potential applications and can be further explored to implement other embedded artificial intelligence tasks, such as visual-language navigation.**
>
> Thank you for highlighting the potential of the simulator for a wide range of applications. Given that Habitat 3.0 is based on the Habitat platform, it automatically supports well-studied Embodied AI tasks [3,4,5]. Furthermore, it opens the opportunity to update many of the existing tasks to include humanoids (e.g., “walk to the human as soon as they sit on the chair”, for visual-language navigation) or to evaluate them using our human-in-the-loop tool.
>
>
> **References.**
>
> [1]  Chen, C., Schissler, C., Garg, S., Kobernik, P., Clegg, A., Calamia, P., ... & Grauman, K. (2022). Soundspaces 2.0: A simulation platform for visual-acoustic learning. Advances in Neural Information Processing Systems (NeurIPS)
>
> [2] Romero, J., Tzionas, D., & Black, M. J. (2017). Embodied hands. ACM Transactions on Graphics, 36(6).
>
> [3] Krantz, J., Lee, S., Malik, J., Batra, D., & Chaplot, D. S. (2022). Instance-Specific Image Goal Navigation: Training Embodied Agents to Find Object Instances. arXiv preprint arXiv:2211.15876.
>
> [4] Szot, A., Clegg, A., Undersander, E., Wijmans, E., Zhao, Y., Turner, J., ... & Batra, D. (2021). Habitat 2.0: Training home assistants to rearrange their habitat. Advances in Neural Information Processing Systems (NeurIPS).
>
> [5] Chaplot, D. S., Gandhi, D. P., Gupta, A., & Salakhutdinov, R. R. (2020). Object goal navigation using goal-oriented semantic exploration. Advances in Neural Information Processing Systems (NeurIPS).

---

### Official Review · Reviewer_ndxG · 2023-10-31

**Soundness:** 2 fair
**Presentation:** 3 good
**Contribution:** 2 fair
**Rating:** 6
**Confidence:** 3

**Summary:**

The paper presents Habitat 3.0, an Embodied AI simulator designed to facilitate research in human-robot interaction and collaboration within complex indoor environments. The proposed platform aims to address the need for efficient simulation tools to study AI agents' capabilities in realistic and diverse human-robot interaction scenarios. The main contributions of the paper are as follows:

Diverse Humanoid Simulation: Habitat 3.0 offers a framework for creating and simulating diverse humanoid avatars. These avatars encompass various appearances and motions, enhancing the realism of agent interactions within the simulated environments. By employing techniques like skeletal models and linear blend skinning, the simulator achieves a balance between efficiency and visual fidelity.

Human-in-the-Loop (HITL) Evaluation Tool: The paper introduces a HITL interface that allows real human operators to control humanoid avatars within the simulated environment. This tool enables online human-robot interaction evaluations and data collection, providing a unique platform for studying how AI agents collaborate with humans.

Social Navigation and Social Rearrangement Tasks: The paper explores two collaborative tasks—social navigation and social rearrangement—to assess AI agents' ability to interact with human or humanoid partners. These tasks require the agents to find and follow humans at a safe distance or to collaborate with a humanoid in rearranging objects within the environment.

Evaluation of AI Agents: The paper compares multiple AI agent baselines in both automated and HITL evaluation settings. These agents include heuristic experts and end-to-end reinforcement learning policies. The evaluations highlight the agents' adaptability in working with different partners to some extent and provide insights into their efficiency and success rates.

Robust HITL Assessments: By conducting HITL evaluations involving real human participants, the paper evaluates AI agents' coordination abilities in scenarios involving diverse partners. These assessments help in understanding how baseline AI agents impact human efficiency and reveal insights into the dynamics of human-robot interactions.

**Strengths:**

Quality:
The paper demonstrates a fair level of quality in its methodology and execution. The development of Habitat 3.0 is well-detailed and addresses a clear need in the field of HRI research. The simulator provides an effective platform for investigating human-robot interaction. The evaluation of AI agents in both automated and HITL settings enhances the paper's quality.

Clarity:
The paper is generally well-written and clear, with detailed explanations of the simulator framework and the tasks studied.

Originality:
Habitat 3.0 introduces valuable original contributions to the field. The combination of diverse humanoid simulation, HITL control, and the study of social navigation and rearrangement tasks provide some initial steps and ideas in this domain. Furthermore, the paper explores HITL evaluations involving human participants, which adds to the originality of the work. While the components themselves are not entirely novel, their integration and application within a single framework is original and significant.

Significance:
Habitat 3.0, with its focus on embodied AI and human-robot interaction, addresses a critical aspect of AI development. The simulator has the potential to open new avenues for research, including collaborative AI, human-robot teamwork, and social embodied agents. The HITL evaluations are particularly significant, offering insights into how AI agents impact human performance and behavior. The paper's findings and methodology are likely to influence future research in these domains.

Overall, its strengths can be listed as:
- focus on human-robot interaction compared to previous platforms offering single-agent or multi-homogeneous-agent training.
- efficient simulation implementation that enables faster progress in training/evaluating developed algorithms.
- human-in-the-loop evaluation tool that can open up interesting use-cases and approaches to improve and analyze HRI methods.
- a fair amount of evaluations.

**Weaknesses:**

The critical weaknesses are:
- currently, the focus of simulations seem to be more on the visual realism, which is a valid concern. However, the movement of the agents lacks physical realism, which hinders the extend of how human-robot interaction can be evaluated accurately.

- the focus of this work is not proposing novel learning algorithms but still the results indicate that none of the baselines achieve useful following rate (F) nor feasible collision rate (CR) in the social navigation task. Similarly, for the social rearrangement task, none of the methods seem to generalize and let the robots assist their partners effectively (checking the relative efficiency (RE) metric). Even for HITL evaluations, which would be simpler since humans adjust to robots on the fly, the results are not encouraging. This, then, makes it harder to evaluate and take some insights from these evaluations, which is a major component of the paper.

- humanoid shape diversity has been considered, however, robotic agent diversity was not addressed.

- similar to the previous point, the platform, as it is now, lacks task diversity as well.

- it feels like (looking at the efficiency improvements (RE metric) when collaborated vs. solo cases) maybe the tasks do not offer enough opportunities for collaboration.

- personally, I find the HITL evaluations more interesting, however, the paper does not cover detailed evaluation and analysis of these experiments.

**Questions:**

- what are possible solutions/integrations to alleviate the unrealistic humanoid locomotion problem, i.e., the agent first rigidly rotates to face the next target waypoint and then follows a straight path. The autonomous agents trained against such human movement models will not be directly transferable to real-world settings, nor the analysis would not be informative.

- it is unclear how easy and flexible to import motion capture data. Can you elaborate on that?

- it is also unclear how trivial it is to use the AMASS dataset along with VPoser to compute humanoid poses and then to import them into the simulator. Trying to use such external tools that the benchmark providers do not support/maintain themselves frequently becomes a huge hassle and ease-of-use of such external tools is critical, so can you also provide some clarification on their integration and/or usability?

- About reliance on VPoser: Depending on the complexity of the task, simple interpolation between poses might not be sufficient, what would be possible solutions?

- is it possible to incorporate physical collaboration scenarios, i.e., partners acting on the same object? would it require additional steps than what was explained on the paper?

---

> ### Author Response · Authors · 2023-11-16
> **Response to reviewer ndxG (Part 1)**
>
> We thank reviewer *ndxG* for the insightful feedback. The questions and concerns are addressed below. The references are in the last part of the response.
>
> **The movement of the agents lacks physical realism, which hinders the extent of how human-robot interaction can be evaluated accurately.**
>
> Habitat 3.0 supports a wide range of humanoid motion, with motion coming from learning-based motion generation models, MoCap data, or by simulating the humanoid dynamically using the Bullet physics backend, such that it responds to forces, gravity, obstacles or momentum. In this work, we choose to simulate the humanoid with some assumptions for efficient simulation, which still achieve a higher level of realism compared to the current Embodied AI simulators with humanoid support, such as VirtualHome [9] or Co-Gail [10].
>
>
> However, there is another deeper question in the reviewer’s comment – how accurate does the simulator need to be? No simulation is perfect, and it is an open question how much physical realism is needed for sim2real transfer. Prior work [11] has shown that dynamic simulation and high fidelity physics might not necessarily lead to better sim2real transfer in navigation. We believe that an analogous systematic sim2real evaluation is necessary for Social Navigation and Social Rearrangement in the context of humanoid simulation to judge the level of realism needed. This is on our roadmap for future work.
>
>
> **The results indicate that none of the baselines achieve a useful following rate (F) nor feasible collision rate (CR) in the social navigation task.**
>
> We believe that only focusing on F alone does not provide an accurate picture of our results. First, note that our RL baseline can almost perfectly locate a moving human in an unseen house, with a success rate of 97%. The Heuristic Expert-designed baseline (which has access to a privileged map of the environment and a path planner), obtains a follow rate of 0.51 and collision rate of 0.52, showing that this is a challenging task, deserving further study. In comparison, the RL baseline achieves a following rate of 0.44, and collision rate of 0.51, without any privileged information. Finally, the following metric (F) is highly influenced by the time it takes to find the human (p). This shows that, while there is room for improvement, our baseline is able to fairly accurately follow the human around cluttered environments. Our experiments establish that these are indeed tasks where state-of-the-art approaches suffer at achieving competitive performance, further enhancing the significance of our work, and opening venues for future research.
>
>
> **For the social rearrangement task, none of the methods seem to generalize and let the robots assist their partners effectively (checking the relative efficiency (RE) metric). Even for HITL evaluations, which would be simpler since humans adjust to robots on the fly, the results are not encouraging.**
>
> While there is room for improvement in our current results, our baselines actually show improvement over performing the task alone. Two agents can, at most, have a relative efficiency of 200%, which would occur if both perfectly split the tasks and did not need to avoid each other (quite unlikely in a typical scenario). Therefore, in both our automated and HITL results, where the relative efficiency is above 100%, our agents effectively assist humans (a maximum of 159.2% and 109.75% for seen and unseen partners, respectively, and 133.80% in the HITL setting). While we agree that human adaptation should further enhance task efficiency in the HITL results, humans also take time to infer their partner's actions, resulting in idle steps that reduce the overall efficiency. The drop in RE for unseen partners further underlines the challenging nature of our tasks, where state-of-the-art approaches have room for improvement.
>
>
> **It feels like (looking at the efficiency improvements (RE metric) when collaborated vs. solo cases maybe the tasks do not offer enough opportunities for collaboration.**
>
> RE is not necessarily a measure of how collaborative a task is since it also measures the ability of a particular approach at improving efficiency over solo execution. In fact, some of our baselines such as Learn-Single (where the agent is trained with a single humanoid policy) offer non-trivial efficiency improvement (160%) when paired with seen partners. This shows that the current social rearrangement task does offer opportunities for collaboration, and merits further study in future work, especially when paired against unseen partners.

---

> ### Author Response · Authors · 2023-11-16
> **Response to reviewer ndxG (Part 2)**
>
> **Robotic agent diversity was not addressed.**
>
> Habitat 3.0 already supports a range of robots (Stretch, Spot, Franka, Fetch) and has the ability to include new robot models (via URDF files) since it is based on the Habitat platform.
>
>
> **Task diversity.**
>
> Since our work builds upon the Habitat platform, it automatically supports social variants of a wide range of Embodied AI tasks, including Image [1], Object [2] or Language [3] based navigation and rearrangement [4], or learning scene representations with embodied agents [5]. In this work, we propose two example tasks that build on navigation and rearrangement, highlighting challenges in human-robot collaboration. However, Habitat 3.0 opens the possibility to define a wide range of tasks, or even update currently studied tasks with humanoid simulation and human-in-the loop interaction. Some ways in which the tasks could be adapted include adding humanoids with complex motions (such as the ones shown in the supplementary video), asking agents to answer questions about the humanoid behavior, anticipating when a humanoid would fall, and performing manipulation tasks in close proximity or jointly with the simulated humanoids such as performing object handovers, or jointly carrying large objects.
>
>
>
> **I find the HITL evaluations more interesting, however, the paper does not cover detailed evaluation and analysis of these experiments.**
>
> Appendix G provides more analysis of the HITL experiments. Specifically, we provide post-hoc pairwise comparisons and significance results between solo execution and the two methods that we evaluate in HITL experiments.
>
>
> **The agent first rigidly rotates to face the next target waypoint and then follows a straight path. What are possible solutions/integrations to alleviate this?**
>
> Two possible solutions to alleviate the simplifying assumption made in our work would be: (1) turning the humanoid before reaching the waypoint in combination with a forward motion, resulting in a smoother curve around the waypoint or (2) training a motion generation model conditioned on waypoints using motion captured data, such as [8].
>
>
> **It is unclear how easy and flexible it is to import motion capture data. Can you elaborate on that?**
>
> Habitat 3.0 represents human motion as a sequence of joint rotations, using the same joints as in the SMPL-X body format. We provide a script to process a SMPL-X pose to be played in Habitat 3.0. As such, any motion capture data can be imported into Habitat by converting it into the SMPL-X format (using open-source tools such as MoSh++ [12]). The script to convert from SMPL-X to the Habitat format can be found in the link below, and will be open sourced.
> https://anonymous.4open.science/r/habitat-lab-C5B4/habitat-lab/habitat/utils/convert_smplx_to_habitat.py
>
> **It is also unclear how trivial it is to use the AMASS dataset along with VPoser to compute humanoid poses and then to import them into the simulator.**
>
> As mentioned above, we provide a script to convert SMPL-X poses into the Habitat 3.0 format. Since VPoser outputs poses in the SMPL-X format, these can directly be played by running the given script, and since AMASS has been processed in the SMPL-X format, it can be directly played as well.

---

> > ### Author Response · Authors · 2023-11-16
> > **Response to reviewer ndxG (Part 3)**
> >
> > **About reliance on VPoser: Depending on the complexity of the task, simple interpolation between poses might not be sufficient, what would be possible solutions?**
> >
> > Indeed, for many humanoid motions, interpolation may not be sufficient. One solution would be to compute novel poses online, using methods that generate motions conditioned on text [6] or 3D scenes [7]. Habitat 3.0 provides support for these methods, however there is a trade-off between the complexity of the motion and the generation speed: fast and less realistic motions are a good fit when simulation speed is a bottleneck (e.g., to train policies with RL) whereas slower higher quality approaches may be preferred in other cases (e.g., to generate datasets offline).
> >
> > **Is it possible to incorporate physical collaboration scenarios, i.e., partners acting on the same object? would it require additional steps than what was explained on the paper?**
> >
> > While physical collaboration scenarios are beyond the scope of this work, Habitat 3.0 readily provides support for them. These scenarios could be enabled by utilizing dynamic simulation, making objects subject to gravity, and potentially unstable when manipulated by a single agent, requiring multi-agent collaboration.
> >
> >
> > **References.**
> >
> > [1] Krantz, J., Lee, S., Malik, J., Batra, D., & Chaplot, D. S. (2022). Instance-Specific Image Goal Navigation: Training Embodied Agents to Find Object Instances. arXiv preprint arXiv:2211.15876.
> >
> > [2] Chaplot, D. S., Gandhi, D. P., Gupta, A., & Salakhutdinov, R. R. (2020). Object goal navigation using goal-oriented semantic exploration. Advances in Neural Information Processing Systems (NeurIPS).
> >
> > [3] Krantz, J., Wijmans, E., Majumdar, A., Batra, D., & Lee, S. (2020). Beyond the nav-graph: Vision-and-language navigation in continuous environments. In European Conference on Computer Vision (ECCV).
> >
> > [4] Szot, A., Clegg, A., Undersander, E., Wijmans, E., Zhao, Y., Turner, J., ... & Batra, D. (2021). Habitat 2.0: Training home assistants to rearrange their habitat. Advances in Neural Information Processing Systems (NeurIPS).
> >
> > [5] Marza, P., Matignon, L., Simonin, O., Batra, D., Wolf, C., & Singh Chaplot, D. (2023). AutoNeRF: Training Implicit Scene Representations with Autonomous Agents. arXiv e-prints, arXiv-2304.
> >
> > [6] Tevet, G., Raab, S., Gordon, B., Shafir, Y., Cohen-or, D., & Bermano, A. H. (2022, September). Human Motion Diffusion Model. In The Eleventh International Conference on Learning Representations (ICLR).
> >
> > [7] Wang, J., Rong, Y., Liu, J., Yan, S., Lin, D., Dai, B. (2022). Towards diverse and natural scene-aware 3d human motion synthesis. In Conference on Computer Vision and Pattern Recognition (CVPR).
> >
> > [8] Hassan, M., Ceylan, D., Villegas, R., Saito, J., Yang, J., Zhou, Y., & Black, M. J. (2021). Stochastic scene-aware motion prediction. In International Conference on Computer Vision (ICCV).
> >
> > [9] Puig, X., Ra, K., Boben, M., Li, J., Wang, T., Fidler, S., & Torralba, A. (2018). Virtualhome: Simulating household activities via programs. In IEEE Conference on Computer Vision and Pattern Recognition (CVPR).
> >
> > [10] Wang, C., Pérez-D’Arpino, C., Xu, D., Fei-Fei, L., Liu, K., & Savarese, S. (2022. Co-gail: Learning diverse strategies for human-robot collaboration. In Conference on Robot Learning (CoRL).
> >
> > [11] Truong, J., Rudolph, M., Yokoyama, N. H., Chernova, S., Batra, D., & Rai, A. (2023). Rethinking sim2real: Lower fidelity simulation leads to higher sim2real transfer in navigation. In Conference on Robot Learning (CoRL).
> >
> > [12] Mahmood, N., Ghorbani, N., Troje, N. F., Pons-Moll, G., & Black, M. J. (2019). AMASS: Archive of motion capture as surface shapes. In Proceedings of the IEEE/CVF international conference on computer vision (pp. 5442-5451).

---

> > > ### Comment · Reviewer_ndxG · 2023-11-22
> > >
> > > Thank you for your explanations. I still think the analyzed task diversity along with some highly simplifying assumptions on the human movement and physical interactions limit the potential of the benchmark in its current form (some of which were also highlighted by other reviewers). Having said that, the study offers a new environment that can pave the way for future improvements. I will re-consider my evaluation based on your clarifications.

---

> > > > ### Author Response · Authors · 2023-11-22
> > > > **Response to reviewer**
> > > >
> > > > Thank you for considering our rebuttal. We are happy to provide further clarifications.

---

### Official Review · Reviewer_FYWe · 2023-11-02

**Soundness:** 3 good
**Presentation:** 3 good
**Contribution:** 3 good
**Rating:** 6
**Confidence:** 4

**Summary:**

- This work introduces Habitat 3.0, a simulation platform for studying human-robot collaboration in home environments.
- The environment includes accurate humanoid simulation and a human-in-the-loop infrastructure for real-time interaction and provides a way to evaluate different robot policies with real human collaborators.
- It also allows the exploration of collaborative tasks: Social Navigation and Social Rearrangement.
- Finally, the authors demonstrate that learned robot policies can effectively collaborate with unseen humanoid agents and human partners. Shows emergent behaviors during task execution, such as yielding space to humanoid agents.

**Strengths:**

1. This work complements existing works (e.g. Habitat, VirtualHome, etc.) in a human-centric way. It allows more flexible human models, human-object interactions, human-robot interactions, and human-in-the-loop evaluation, which are often ignored in previous works.
2. This work is in general well-written, providing a good survey for the field of embodied AI environments.
3. This work designs two social/collaborative tasks, navigation and rearrangement, and shows promising results.

**Weaknesses:**

The main limitation of this work lies in universality. While I believe this work is interesting and helpful to the field, I am wondering if it could be scaled to incorporate more elements, supporting more tasks, so that the progress wouldn't stop here at the two example tasks. While I understand that these aspects might be beyond the scope of a single work, it would be beneficial to demonstrate, or at least discuss, how future works can develop upon Habitat 3.0. For example,
- physics simulation
- fine-grained object interactions
- sim2real deployment

**Questions:**

1. For social arrangement, what is the motivation for using population-based approaches? More discussions would be helpful to understand the setting.
2. Discuss relevant previous works. What is the relationship between the proposed social navigation task and visual tracking (e.g. [a])? Seems quite similar and more discussions are needed. Besides, [a] also contains humanoid simulation and end-to-end RL with Conv+LSTM.
3. For object interaction, Sec. 3.1 explains that "Once the hand reaches the object position, we kinematically attach or detach the object to or from the hand." Are all the objects simplified as a point particle? Are all the objects in the environment interactive? Is it possible to add more properties to them (e.g. geometry - shape, physics - weight, material, etc.)? It would be great to explore/discuss how to incorporate more general and complex object interactions, from pre-defined motion primitives (e.g. lie, sit) to freeform actions (e.g. grasp).
4. Currently this work focuses restricted set of motions while more motions can be potentially added with the SMPL-X representation. In demo video 2:56, it also discusses complex motions (e.g. wiping, knocking). It would be beneficial to discuss how future works can incorporate more motions with interactive objects and form meaningful tasks.
5. Another question is, what kind of tasks can Habitat 3.0 support in addition to navigation and arrangement? Again, I understand that these two tasks are already great for this work, but more discussions on the potential of Habitat 3.0 would make this work more general and influential.
6. I wonder if the HITL tools would also be standardized and open-source.

[a] End-to-end Active Object Tracking via Reinforcement Learning, ICML 2018.

---

> ### Author Response · Authors · 2023-11-16
> **Response to reviewer FYWe (Part 1)**
>
> We thank reviewer *FYWe* for the insightful feedback. The questions and concerns are addressed below. The references are in the last part of the response.
>
> **Universality in terms of (1) physics simulation, (2) sim2real deployment, (3) fine-grained object interaction.**
>
> Note that Habitat 3.0 is built on top of the Habitat platform, providing support for physics simulation, a wide range of tasks and examples of sim2real transfer. We provide more details below:
>
> - Physics simulation: Habitat 3.0 supports physics simulation via bullet, allowing to specify physical properties for every object or agent and to simulate rigid-body dynamics (forces, collisions, etc). Our current work primarily uses a ‘kinematic’ simulation, which does not model dynamics. Our future work will include more sophisticated physical simulation for human-robot interaction, such as joint object manipulation and handovers.
>
> - Sim2Real deployment: Previous works leveraging the Habitat platform have established a track record of sim2real deployments in the context of navigation [1] and mobile manipulation [2]. Since Habitat 3.0 uses the same physics and rendering engine, and all policies in our submission use depth-sensors - which have shown to have low sim2real gap [1,3] - we believe that there is potential for sim2real deployment in the tasks we study. We leave this as a venue for future work.
>
> - Fine Grained Object Interaction: In this work, our focus was primarily on high-level interactions such as task division and reducing space interference with the other agent. In our future work, we will consider more complex interactions such as lying and sitting for our tasks. However, the inclusion of features such as dexterous manipulation remains outside the current scope of Habitat 3.0. We believe other simulators and physics engines (e.g., IsaacSim) are particularly well-suited and designed for these tasks.
>
>
>
> **Universality in terms of tasks.**
>
> We propose social variants of two commonly studied tasks in Embodied AI simulators (i.e. navigation and rearrangement), adapting them to human-robot collaboration scenarios. Similarly, we can create social variants of other embodied AI tasks, such as embodied question answering [10], object [7], image [8] or language navigation/rearrangement [9] by incorporating the humanoids. Furthermore, the simulator can be potentially used for other human-centered tasks, such as long-term scene-dependent motion generation with language input, anticipation of human actions, and human intent inference since it enables generating motions for humanoids and it includes the human-in-the-loop infrastructure to evaluate the tasks with real people.
>
>
> **For social rearrangement, what is the motivation for using population-based approaches?**
>
> The goal in social rearrangement is to build robot agents that can rearrange objects together with humans, who can perform this task in different ways. An effective robot agent needs to adapt to its partner’s preferences to effectively assist them. We frame this as a zero-shot coordination problem and refer to [6], which has demonstrated that training agents with a diverse population enables this type of adaptation. Note that we also present a baseline (Learn-Single) which does not use a population, and instead learns to coordinate with a single partner. Our experiments show that this baseline performs worse than population-based approaches at zero-shot coordination with unseen partners (in both automated and human-in-the-loop settings).
>
> **Relationship between Active Object Tracking and Social Navigation.**
>
> Thank you for the reference; we have updated the paper with the relationship of our task with Luo et al. [5]. We note two important differences between [5] and the SocialNav task: First, in [5] there is no robot to control, and thus, does not consider important human-robot interaction aspects that we observe in our work, such as being aware of the robot’s physical extent, avoiding human-robot collisions by maintaining a safe distance or backing up when necessary to unblock the human. Second, [5] assumes that the human to track is mostly in the field of view of the robot, whereas in our case, the robot starts far away from the human and thus has to first explore the environment to find the human (Find phase) before following and tracking it around the environment (Follow phase).

---

> ### Author Response · Authors · 2023-11-16
> **Response to reviewer FYWe (Part 2)**
>
> **Clarification for Section 3.1. Are all the objects simplified as a point particle? Are all the objects in the environment interactive? Adding properties to objects.**
>
> Objects are *not* represented as point particles; instead, they have volumetric collision meshes and physical properties, which can be edited by researchers. Some objects can also be articulated allowing agents to open doors or cabinets. For the experiments in this work, we do not simulate dynamics (forces, momentum, friction, etc.) and kinematically attach the object volume to the agent arm, but the simulator allows for dynamic simulation. We have clarified this in the updated revision.
>
>
> **It would be beneficial to discuss how future works can incorporate more motions with interactive objects and form meaningful tasks.**
>
> Our work provides an interface to represent human poses or motions via the SMPL-X format (i.e. as a sequence of rotations for each human joint). As such, any model or system that can generate these poses, could also be used in Habitat 3.0. Researchers could for example use [4], to generate human motion from language descriptions, allowing for tasks that require robots to reason about these motions (for instance, wait for a person to sit in the chair before bringing them a glass of water).
>
>
>
> **I wonder if the HITL tools would also be standardized and open-source.**
>
> Yes, we will open-source the HITL tool, together with the simulator and experiment code. The source code for the HITL tool can be found in the following anonymous link: https://anonymous.4open.science/r/habitat-lab-C5B4/README.md.
>
>
> **References.**
>
> [1] Truong, J., Chernova, S., & Batra, D. (2021). Bi-directional domain adaptation for sim2real transfer of embodied navigation agents. IEEE Robotics and Automation Letters, 6(2), 2634-2641.
>
> [2] Yokoyama, N., Clegg, A. W., Undersander, E., Ha, S., Batra, D., & Rai, A. (2023). Adaptive Skill Coordination for Robotic Mobile Manipulation. arXiv e-prints.
>
> [3] Truong, J., Rudolph, M., Yokoyama, N. H., Chernova, S., Batra, D., & Rai, A. (2023). Rethinking sim2real: Lower fidelity simulation leads to higher sim2real transfer in navigation. In Conference on Robot Learning (CoRL).
>
> [4] Tevet, G., Raab, S., Gordon, B., Shafir, Y., Cohen-or, D., & Bermano, A. H. (2022). Human Motion Diffusion Model. In International Conference on Learning Representations (ICLR).
>
> [5] Luo, W., Sun, P., Zhong, F., Liu, W., Zhang, T., & Wang, Y. (2018). End-to-end active object tracking via reinforcement learning. In International Conference on Machine Learning (ICML).
>
> [6] Szot, A., Jain, U., Batra, D., Kira, Z., Desai, R., & Rai, A. (2023). Adaptive Coordination in Social Embodied Rearrangement. In International Conference on Machine Learning (ICML).
>
> [7] Chaplot, D. S., Gandhi, D. P., Gupta, A., & Salakhutdinov, R. R. (2020). Object goal navigation using goal-oriented semantic exploration. Advances in Neural Information Processing Systems (NeurIPS).
>
> [8] Krantz, J., Lee, S., Malik, J., Batra, D., & Chaplot, D. S. (2022). Instance-Specific Image Goal Navigation: Training Embodied Agents to Find Object Instances. arXiv preprint arXiv:2211.15876.
>
> [9] Krantz, J., Wijmans, E., Majumdar, A., Batra, D., & Lee, S. (2020). Beyond the nav-graph: Vision-and-language navigation in continuous environments. In European Conference on Computer Vision (ECCV).
>
> [10] Das, A., Datta, S., Gkioxari, G., Lee, S., Parikh, D., & Batra, D. (2018). Embodied question answering. In IEEE Conference on Computer Vision and Pattern Recognition (CVPR).

---

> ### Comment · Reviewer_FYWe · 2023-11-22
> **Further discussions**
>
> Thank the authors for the detailed response, and sorry for the delay in response. My remaining questions are as follows:
> 1. Given that the Habitat environment is bullet-based and naturally supports physics simulation, why does this work primarily consider kinematic simulation? Is this due to the focus of current human parametric models, such as SMPL, on kinematics without incorporating physics?
> 2. Currently the object interactions are somewhat simple, and do not make full use of the interactive objects (e.g. doors or cabinets). These objects can make the proposed tasks (and other potential ones) more interesting. For instance, in virtualhome people can open the cabinet and search for objects. Is it because combining these object state changes with realistic human motions is more challenging?
> 3. Compared to existing projects like VirtualHome, this work seems to allow greater freedom in movement, such as freeform walking, but offers more limited action types and object interactions (limited to walking and pick/place actions). Given that VirtualHome is more symbolic and rule-based, I am curious if we can merge these advantages to enhance the versatility of human-centric virtual environments.
>
> I appreciate the authors' discussion and insights on these questions.

---

> > ### Author Response · Authors · 2023-11-22
> > **Response on further discussion**
> >
> > Thank you for your response. We address the remaining questions below.
> >
> >
> > **Given that the Habitat environment is bullet-based and naturally supports physics simulation, why does this work primarily consider kinematic simulation? Is this due to the focus of current human parametric models, such as SMPL, on kinematics without incorporating physics?**
> >
> > Our choice is primarily based on prior work [3], which has shown that policies trained with kinematic simulation outperform those trained with dynamic simulation when transferred to the real world since they allow for faster training and provide a more abstract representation that can generalize better across domains. Our humanoid avatars are rigged, and can therefore be dynamically simulated using works such as [11]. However, robust dynamic humanoid control is still an open research question, and we leave working with dynamically controlled humanoids for future work. Additionally, simulation speed is crucial for us, as our algorithms require on the order of hundreds of millions of iterations to converge. Kinematic simulation is much more efficient compared to dynamic simulation, though it comes at the expense of slightly less realism.
> >
> >
> > **Currently the object interactions are somewhat simple, and do not make full use of the interactive objects (e.g. doors or cabinets). These objects can make the proposed tasks (and other potential ones) more interesting. For instance, in virtualhome people can open the cabinet and search for objects. Is it because combining these object state changes with realistic human motions is more challenging?**
> >
> > Habitat 3.0 supports these kinds of object interactions, and prior work [12] has shown success in training robot policies to perform them. VirtualHome implements these actions by setting the human hand to follow a predefined per-object trajectory, so that it follows the handle of a cabinet when it opens, and animating the rest of the body via inverse kinematics (IK). In some cases this leads to artifacts (see the opening cabinet motion on the top right video here: https://youtu.be/lrB4K2i8xPI?si=O-Hfa8vpwkcnHLDK&t=78 time 1:18). A key advantage in our work is the flexibility of our representation, allowing us to use the above IK-based solution (via the VPoser method described in the paper) but also include more sophisticated learning-based methods trained on motion capture data [13].
> >
> >
> > We agree that including these interactions in our tasks would add extra complexity, and the next version of the framework includes such interactions. However, we want to emphasize that even in the current setting, there are still challenges and opportunities to learn more sophisticated agent behaviors which more effectively split tasks with humans (as noted by the gap between the training and eval populations) or better operate without interfering with humans actions.
> >
> >
> >
> >
> > **Compared to existing projects like VirtualHome, this work seems to allow greater freedom in movement, such as freeform walking, but offers more limited action types and object interactions (limited to walking and pick/place actions). Given that VirtualHome is more symbolic and rule-based, I am curious if we can merge these advantages to enhance the versatility of human-centric virtual environments.**
> >
> >
> > This is a great suggestion, and it is something we are actively working on. We want to emphasize that our current scope already presents several interesting challenges, but note that Habitat can enable merging these two domains, as suggested, where high level policies or planners could call the policies learned in our work. We identify two key elements to enable these richer interactions and object states:
> >
> >
> > First, Habitat 3.0 supports a wider range of interactions (including opening cabinets or drawers). Prior work [12] has successfully deployed these interactions in Habitat, using robot embodiments, and these could be included in humans by using the methods described in the previous response. Similarly, some of the more sophisticated human motions in VirtualHome (e.g. drinking or sitting) could also be added by using existing motion capture data or learning-based models [14] compatible with our representation.
> >
> >
> > Second, since Habitat allows us to query the geometry of all objects in the environment, we can parse this information to obtain a symbolic representation of the scene. Such representation would describe object states (closed, open) and relationships between objects (e.g. inside or on top) which would be updated by agent actions or dynamic simulation forces (e.g. an object on the edge of the table can fall on the floor).

---

> > > ### Author Response · Authors · 2023-11-22
> > > **Response on further discussion - References**
> > >
> > > **References**
> > >
> > > [11] ​​Luo, Z., Yuan, Y., & Kitani, K. M. (2022). From Universal Humanoid Control to Automatic Physically Valid Character Creation. arXiv e-prints, arXiv-2206
> > >
> > > [12] Szot, A., Clegg, A., Undersander, E., Wijmans, E., Zhao, Y., Turner, J., ... & Batra, D. (2021). Habitat 2.0: Training home assistants to rearrange their habitat. Advances in Neural Information Processing Systems, 34, 251-266.
> > >
> > > [13] Pons-Moll, G., Guzov, V., Chibane, J., Marin, R., He, Y., & Sattler, T. In International Conference on 3D Vision (2024). Interaction Replica: Tracking human-object interaction and scene changes from human motion.
> > >
> > > [14] Hassan, M., Ceylan, D., Villegas, R., Saito, J., Yang, J., Zhou, Y., & Black, M. J. (2021). Stochastic scene-aware motion prediction. In Proceedings of the IEEE/CVF International Conference on Computer Vision (pp. 11374-11384).

---

### Author Response · Authors · 2023-11-16
**General Response - Revisions**

We appreciate the insightful comments from the reviewers. We are encouraged by their acknowledgment that Habitat 3.0 advances research in human-robot interaction (FYWe, ndxG, HoUA), provides insights through the human-in-the-loop evaluation (ndxG, HoUA), and that the paper is clear and thorough (FYWe, ndxG, HoUA).

We have added the following revisions to the paper (shown in dark blue):

-  Further clarified the general abilities of our simulator, including diverse motions and tasks, beyond those presented in the paper (Appendix F.2).
- Clarified that our work builds upon the Habitat platform, supporting multiple tasks, simulation features, and embodiments (Appendix F.2).
-  Added a reference to Luo et al., ICML 2018, in the context of social navigation (Sec 4.1).
-  Clarified the physics and object interaction details in the paper (Appendix F.2).

We also provide the source code for the human-in-the-loop tool at this link (“hitl” folder): https://anonymous.4open.science/r/habitat-lab-C5B4/README.md, together with a script to convert SMPL-X poses, and the experiment code shared in the original submission.

We provide more details in the direct responses to the reviewers.

---

### Author Response · Authors · 2023-11-20
**Note to Reviewers - Additional Feedback**

Dear Reviewers,

We hope that our response has effectively addressed your comments and concerns. As we approach the deadline for the discussion period, we wanted to kindly ask you if you could review our response. We would greatly appreciate if you could let us know of further questions, and if you would consider updating your scores given our response.

---

### Meta-Review · Area_Chair_Lra8 · 2023-12-06

**Metareview:**

This paper proposes a practical simulation platform involving humans and robots. Specifically, it simulates humanoid motion (e.g., walking towards a target position) and collaborative tasks, with a human-in-the-loop infrastructure. The paper demonstrates the effectiveness of the simulator on two tasks: Social Navigation and Social Rearrangement, showing impressive results compared against baseline methods. Reviewers recognize that this work compensates existing platforms by providing human-centric simulations, thus potentially has a impact in relevant research areas. The paper is also well-written with sufficient technical details. For weaknesses, reviewers suggested further exploration for more tasks, and modeling diverse human motions, including dexterous interactions with objects.

**Justification For Why Not Higher Score:**

This paper proposes a practical platform for human-centric simulation. It potentially provide an effective platform for relevant research areas. However, limitations such as task and human motion diversity, especially for complex human-object interactions exist, thus I recommend to accept the paper as a poster representation.

**Justification For Why Not Lower Score:**

All reviewers have position feedbacks of the paper for its potential impact in relevant research area. Most concerns on the details have been adequately addressed during the rebuttal. While some limitations are acknowledged, they do not diminish the overall contribution of the paper. Thus I recommend to accept it.

---

### Decision · Program_Chairs · 2024-01-16

Accept (poster)